# Food purchase patterns in Nairobi before, during, and after the COVID-19 pandemic lockdown measures

**Reinpeter Momanyi**[1]*, **Tatenda Duncan Kavu**[1,2], **Daniel Mtai Mwanga**[1,3], **Caroline H. Karugu**[1,4], **Steve Cygu**[1], **Gershim Asiki**[1], **Agnes Kiragga**[1]

**1** African Population and Health Research Center, Nairobi, Kenya, **2** Department of Computer Science, College of Science, Engineering and Technology, University of South Africa, South Africa, **3** Department of Mathematics, University of Nairobi, Nairobi, Kenya, **4** Department of Public and Occupational Health, Amsterdam Public Health, University of Amsterdam Medical Centers, Amsterdam, the Netherlands

* rmomanyi@aphrc.org, reinp.mom@gmail.com

## Abstract

The nationwide lockdown measures implemented during the coronavirus disease 2019 (COVID-19) pandemic disrupted food supply systems, potentially altering consumer purchasing behaviour. There is limited evidence quantifying these changes in low- and middle-income settings. This study aimed to examine the impact of COVID-19 on grocery purchase patterns among consumers in Nairobi. Using generalized least squares (GLS), we conducted an interrupted time series (ITS) analysis of weekly food purchase data from 2018 to 2023. The analysis considered three periods: pre-COVID (12th January 2018–26th March 2020), COVID (27th March 2020–21st October 2021), and post-COVID (22nd October 2021–31st December 2023). A total of 11,105,974 transactions from two supermarkets in Nairobi were classified using the NOVA food classification and linked with nutrient composition data. Compared to the pre-COVID period, characterized by declining purchases of processed culinary, unprocessed/minimally processed foods, and increasing ultra-processed food (UPF) purchases, the COVID period was associated with a short-term change toward healthier purchasing patterns, including reduced UPF, with increased processed and unprocessed/minimally processed food purchases. Nutritionally, pre-COVID trends of rising energy and carbohydrate purchases and declining proteins, calcium, iron, magnesium, potassium, and sodium, were contrasted by short-term increases during COVID in fibre, iron, magnesium, phosphorus, potassium, and sodium, alongside a long-term decline in carbohydrates. Proteins showed consistent short- and long-term increases, while calcium rose sharply at COVID onset but declined over time. In comparison, the post-COVID period reflected a reversal of these changes. While processed food purchases increased briefly before declining, longer-term trends showed increases in calcium, sodium, and carbohydrate purchases and decreases in energy and fat. These findings shed light on how populations adapt their food purchasing

**Data availability statement:** The datasets generated during and/or analysed during the current study are publicly available in the APHRC Microdata Portal Repository via (https://microdataportal.aphrc.org/index.php/catalog/233). The source code used to produce the results and analyses presented in this manuscript is available from GitHub (https://github.com/reinpmomz/Impact-of-COVID_19-on-Food-Purchase-Patterns-Using-ITS).

**Funding:** This research was supported by in-house seed funds from the African Population and Health Research Center (APHRC) to AK. The funders had no role in study design, data collection and analysis, decision to publish, or preparation of the manuscript.

**Competing interests:** The authors have declared that no competing interests exist.

behaviors during and after global crises, offering insights that can inform future policies aimed at curbing unhealthy food purchases and strengthening food security.

## 1. Introduction

Following the emergence of COVID-19 caused by the novel severe acute respiratory syndrome coronavirus 2 (SARS-CoV-2) in December 2019 [1], Kenya, like many other countries, was significantly affected by a series of COVID-19 waves and circulation of SARS-CoV-2 variants, which severely disrupted social, public, and individual life [2]. Within Kenya, Nairobi consistently experienced the highest burden of confirmed COVID-19 cases throughout the pandemic. The city's central role in economic activity and inter-county transportation, its high population density, and reliance on daily population movement heightened both exposure risk and vulnerability to transmission [3].

As part of mitigation measures to prevent the spread of COVID-19, the Kenyan Government announced the closure of all educational institutions and banned public gatherings aimed at minimising transmission on 15th March 2020 [4]. A nationwide dusk-to-dawn curfew was implemented on 27th March 2020, which included the closure of all but 'essential' businesses, including full-service restaurants and other out-of-home (OOH) food establishments [5]. While necessary for infection control, these measures had pronounced economic consequences in Nairobi, where a large proportion of households rely on informal sector incomes that were immediately disrupted [6,7]. The lockdown adversely affected the food supply chain, transport, food security, healthcare services, employment, social interactions, and income levels [6,7]. Additionally, there was a significant impact on health and consumer behaviors, including changes in daily routines, sleep, exercise, sedentary behavior, diet, and food purchase patterns [8,9].

During the lockdown period, there were restrictions on mobility that led to a noticeable change in consumer behavior. Grocery stores and supermarkets became the main places to shop for food, particularly in urban centers such as Nairobi where modern retail outlets are more accessible [10]. Food shopping patterns shifted to fewer trips but with more spending, with people stockpiling food items, including frozen foods, out of fear that they may not have enough food reserves, but more importantly, to minimize exposure risk [11–13]. As the pandemic progressed, additional restrictions rendered physical store shopping disadvantageous. In-store shopping was not only viewed as high risk, but it was also tough to visit a single store and find it fully supplied due to supply chain disruptions. As a result, customers began utilizing online grocery food shopping platforms and home delivery services [14,15].

Pandemic-related restrictions led to both health-promoting and health-damaging changes [16–18]. Negative changes included consumption of unhealthy foods, such as salty snacks, sweet snacks, sugary drinks, and processed foods [18–20]. Diverging observations have been made in different populations, with people making improvements in their diet quality during the COVID-19 pandemic. Studies conducted in Bangladesh, Hong Kong, Italy, Russia, and the USA [11,12,15,21,22] reveal that the availability of time, concerns regarding the potential risks of dining out, and

income disparities significantly contributed to the increase in the preparation of meals at home. This change toward cooking at home allowed individuals to hone their culinary skills, transition toward healthier food choices, and foster better food management practices by reducing food wastage.

In Kenya, the nationwide income reductions due to the COVID-19 pandemic [6,7] led to diminishing purchasing power, with households opting to consume simple foods, such as maize, beans, potatoes, fruits, and vegetables [23,24]. These effects were more pronounced in Nairobi, where job losses and reduced earnings significantly constrained household food budgets. Furthermore, supply chain disruptions in Nairobi led to significant rise in staple food prices including rice by 11.3%, lentils and pulses (beans, peas, green grams) by 11.3 – 19.4%, cereal grains (maize, wheat, sorghum) by 12.8 – 14.1%, meat and poultry (beef, mutton, chicken) by 13.9%, vegetables by 25.1%, fruits by 14.4%, milk and milk products by 15.3% and other food items (cooking oil, spices, sugar) by 26.2% [23,25]. The rising food prices in Nairobi's urban markets further compounded food access challenges, which forced households to reduce the number of meals eaten in a day, reduce food portions, reduce dietary diversity [23,25], and opt for cheaper, less nutritious foods [17]. Similar patterns have been observed across African contexts, particularly in rapidly urbanizing cities. Studies done in Burkina Faso, Ethiopia, Nigeria, Tanzania, Rwanda, Sierra Leone and Ghana have also reported reduced dietary diversity, increased food insecurity, and shifts toward cheaper, energy-dense foods during the lockdown period [26–28].

Beginning late 2021 onwards, many countries entered the endemic stage of the COVID-19 outbreak and started transitioning from restrictions to targeted vaccination for managing the endemic disease. On 20th October 2021, the Kenyan Government ended the dusk-to-dawn curfew that had been in effect since 27th March 2020 [29]. Although the pandemic accelerated the trend towards searching for healthier and sustainable food products, it was anticipated that dietary and purchasing habits practiced during confinement would differ during the post-confinement period.

Although studies have reported changes in shopping habits globally during [12,20,21] and after [10,19] the COVID-19 pandemic, evidence from sub-Saharan Africa remains limited, particularly using longitudinal purchase data. Most existing studies lack the temporal granularity needed to distinguish short-term disruptions from sustained behavioral change. This study addresses these gaps by leveraging, large-scale transaction-level supermarket purchase data from two supermarkets in Nairobi to examine changes in food purchasing and nutritional composition in an urban setting in Kenya before, during, and after the COVID-19 pandemic. The supermarkets were selected based on data availability and willingness to share transaction-level data. While the sample was not designed to be statistically representative of all retail outlets in Nairobi, the selected supermarkets serve a large and diverse urban customer base and provide a stable platform for examining temporal changes in purchasing patterns.

By applying an interrupted time series framework that captures both the onset and offset of COVID-19 pandemic restrictions, this paper generates robust evidence on immediate and long-term purchase behavior to a major public health shock. Understanding these dynamics in an urban African context is critical for informing nutrition policy, food system resilience strategies, and public health interventions aimed at mitigating the adverse impacts of future shocks.

## 2. Materials and methods

### 2.1 Ethics statement

Ethical approval was granted by the Amref Ethics and Review Committee (ESRC) in Kenya (IRB number AMREF-ESRC P1526/2023). Approval to conduct the research was granted by the National Commission for Science, Technology & Innovation (NACOSTI) in Kenya (License number NACOSTI/P/24/32249). As this study utilized secondary, anonymized data, no direct participant contact occurred.

### 2.2 Data source and study setting

De-identified item-level transaction data on food purchases were obtained through secondary data collection from electronic transactional records of two supermarket chains in Nairobi County, Kenya. These electronic records comprised

point-of-sale (POS) purchase information collected from 12th January 2018–31st December 2023. Each transaction record included a unique transaction ID, anonymized supermarket identifier, product description, purchase quantity, price, transaction timestamp, and supermarket location. The data were made available to the research team under a formal data-sharing agreement.

Food composition data were primarily obtained from the Kenya Food Composition Tables 2018 (KFCT), developed by the Ministry of Health and the Ministry of Agriculture and Irrigation with support from the Food and Agriculture Organization of the United Nations (FAO) [30]. Food composition data for food items that were not found in the KFCT were secondarily obtained from the Composition of Foods Integrated Dataset (CoFID) [31,32] and the Food Composition Tables for use in the English-speaking Caribbean [33].

## 2.3 Data abstraction and processing

Data abstraction involved securely transferring raw transactional data with the main variables of interest into a controlled research environment. Only anonymized records were accessed, with all direct personal identifiers removed prior to data transfer. The extracted grocery data in the form of CSV files was saved into a local database using PostgreSQL version 15.2 [34]. A reproducible extraction, transformation, and loading (ETL) pipeline was developed using R version 4.5.0 [35] to facilitate product categorization, harmonization, and data cleaning. The raw data had 11,229,879 transactional records. Data pre-processing techniques were applied to check for data consistency and quality assurance checks, and filtering of non-food items. The final dataset for analysis had 11,105,974 transactional records. Detailed information about the data processing workflow has been published [36].

Following the data pre-processing step, food items were classified according to the NOVA food classification system [37], which categorizes foods based on the extent and purpose of industrial processing into four groups: unprocessed or minimally processed foods, processed culinary ingredients, processed foods, and ultra-processed foods. All food items were classified by two reviewers using published NOVA guidelines and adapted to the local food context. Product descriptions, and where available, brand information were used to assign each item to a NOVA category. A classification codebook was maintained to ensure transparency and reproducibility. Food items were then subsequently linked to corresponding food composition data. Further details on the NOVA food classification are provided in S1 Appendix.

## 2.4 Study design

This study uses an Interrupted Time Series (ITS) design to estimate changes in food purchase patterns following the start and end of COVID-19 pandemic restrictions. Interrupted time series is an analysis method proposed by Box and Tiao to evaluate the impact of intervention measures on outcomes by comparing observed post-event outcomes with those calculated by continuing the trend observed before the event, that is, the counterfactual [38,39]. ITS controls the original regression trend of outcomes before and after the intervention, compares the immediate level changes of outcomes, and evaluates the impact of intervention measures on outcomes in short-term and long-term dimensions. Interrupted time series is a type of quasi-experimental design used to evaluate the impact of interventions on longitudinal data, particularly in settings where interventions occur as natural experiments without a formal control group [39]. We specified the times of the intervention as 27th March 2020 (first interruption) and 20th October 2021 (second interruption), when the dusk-to-dawn curfew was implemented and ended, respectively, in Kenya. Correspondingly, our study period consisted of a total of 312 weeks divided into 115 pre-COVID (12th January 2018–26th March 2020), 82 COVID (27th March 2020–21st October 2021), and 115 post-COVID (22nd October 2021–31st December 2023) weeks.

## 2.5 Purchase outcomes

Purchase data were aggregated to weekly intervals prior to analysis. Weekly aggregation reduced high-frequency noise variability due to day-of-week seasonality and autocorrelation while preserving meaningful temporal variation and aligning with typical household grocery shopping cycles and retail promotion schedules. This level of aggregation also provided a

balance between temporal granularity and statistical stability, ensuring sufficient time points for robust estimation of pre- and post-intervention trends in the interrupted time series model [39,40]. We considered the following outcomes:

i. Proportion of food purchases according to four NOVA food categories (unprocessed foods, processed culinary ingredients, processed foods, and ultra-processed foods).

ii. Mean proximate nutrient composition values per 100g/100ml of food, i.e., energy (kcal), water (g), protein (g), fat (g), carbohydrate (g), fibre (g), and cholesterol (mg).

iii. Mean mineral nutrient composition values per 100g/100ml of food, i.e., calcium (mg), iron (mg), magnesium (mg), phosphorus (mg), potassium (mg), sodium (mg), zinc (mg), and selenium (mcg).

iv. Mean vitamin nutrient composition values per 100g/100ml of food, i.e., vitamin A-RE (mcg), thiamin (mg), riboflavin (mg), niacin (mg), dietary folate equivalent (mcg), vitamin B12 (mcg), and vitamin C (mg).

## 2.6 Statistical analysis

Descriptive summary statistics in the form of frequency, percentage, mean, standard deviation (SD), median, interquartile range (IQR), and range were done to examine the food purchases and nutrient composition values. Autocorrelation was assessed by using the autocorrelation function (ACF) and Durbin-Watson test statistics, which showed serial correlation in all outcome variables. We also used the augmented Dickey-Fuller unit root test and the Kwiatkowski-Phillips-Schmidt-Shin (KPSS) trend-stationary test to determine the nature of the trends. Both tests showed that the majority of the trends were stationary and non-deterministic; hence, the generalized least squares (GLS) model was the recommended approach for the ITS analysis. Unlike traditional models, which focus primarily on forecasting through differencing and lag structures, the GLS model provides a robust and flexible framework for estimating time-dependent intervention effects by efficiently accounting for autocorrelation and heteroscedasticity in the error terms through transforming the model and using ordinary least squares (OLS) on the adjusted data [41,42]. Autoregressive and moving average (ARMA) orders (p and q) for the GLS model were determined via a grid search. Serial autocorrelation in the residuals was accounted for by specifying candidate ARMA(p,q) correlation structures with p and q ranging from 0 to 4, and the fitted model with the lowest AIC was selected. Separate ITS models were fitted for multiple outcomes to estimate level and trend changes following the onset and offset of COVID-19 restrictions. Outcomes were specified a priori and organised into conceptual groupings relevant to dietary quality, including proportions of purchases by NOVA category and mean nutrient composition measures (proximate, minerals, and vitamins) per 100 g/100 ml of food. Interpretation focused on statistical significance, magnitude, and direction of effect sizes. Formal adjustment for multiple testing was therefore not applied. Statistical significance was set at p-value <0.05, meaning coefficients with p-values below this threshold were considered statistically significant. Model fitness and performance were evaluated using several post-estimation metrics, including Akaike Information Criterion (AIC), Bayesian Information Criterion (BIC), Root Mean Squared Error (RMSE), Mean Absolute Error (MAE), Mean Absolute Percentage Error (MAPE), and Mean Absolute Scaled Error (MASE). All the analysis and data visualisation were carried out using R version 4.5.0 [35].

## 2.7 Model equation

This study focused on temporal changes in purchasing behavior. The ITS model included terms for baseline level and trend, as well as immediate level change for capturing short-term behavioral responses and slope changes for capturing sustained behavioral adjustments associated with the onset and lifting of COVID-19 restrictions. No additional time-varying covariates were included as the models did not adjust for temporal food price variation. The standard ITS analysis GLS model to assess the impact of the two interventions assumed the form as indicated by equation (1).

$$Y_t = \beta_0 + \beta_1 T + \beta_2 I_1 + \beta_3 P_1 + \beta_4 I_2 + \beta_5 P_2 + \varepsilon_t \tag{1}$$

where $Y_t$ is the outcome variable at time $t$, $T$ is a continuous variable indicating time (in weeks) at time $t$ from the start ($t = 1$ week) until the end ($t = 312$ weeks) of the observation period (January 2018 to December 2023), $I_1$ is a dummy variable indicating the first interruption and is coded 0 for time occurring up to the last point before the first intervention ($t = 1$–115 weeks) and 1 for time occurring on and after the first intervention ($t = 116$–312 weeks), $P_1$ is a variable indicating time passed since the first intervention and is coded 0 for time occurring up to the last point before the first intervention ($t = 1$–115 weeks) and 1–197 after the first intervention ($t = 116$–312 weeks), $I_2$ is a dummy variable indicating the second interruption and is coded 0 for time occurring up to the last point before the second intervention ($t = 1$–197 weeks) and 1 for time occurring on and after the second intervention ($t = 198$–312 weeks), $P_2$ is a variable indicating time passed since the second intervention and is coded 0 for time occurring up to the last point before the second intervention ($t = 1$–197 weeks) and 1–115 after the second intervention ($t = 198$–312 weeks) and $\varepsilon_t$ is the error term at time $t$. Similarly, $\beta_0$ represents the baseline value of the outcome (the intercept or constant) at time zero, $\beta_1$ is the slope (change over time) before any intervention was implemented, $\beta_2$ represents the immediate change in the outcome measure from the last time point before the first intervention to the first time point after the first intervention, $\beta_3$ is difference in the slope of the period before the first intervention and the slope of the period after the first intervention, $\beta_4$ represents the immediate change in the outcome measure from the last time point before the second intervention to the first time point after the second intervention and $\beta_5$ is the difference in the slope of the period before the second intervention and the slope of the period after the second intervention.

In the absence of the first and second interventions, the GLS model predicts the outcomes in the form given by equation (2).

$$Y_t = \beta_0 + \beta_1 T + \varepsilon_t \tag{2}$$

In the absence of the second intervention, the GLS model predicts the outcomes in the form given by equation (3).

$$Y_t = \beta_0 + \beta_1 T + \beta_2 I_1 + \beta_3 P_1 + \varepsilon_t \tag{3}$$

Equations (2) and (3) calculate the counterfactual (what would have occurred to the outcome, had the interventions not happened).

### 2.8 Sensitivity analysis

We repeated the ITS analysis using the Autoregressive Integrated Moving Average (ARIMA) model. The ARIMA model is one of a series of time series analysis methods proposed by Box and Jenkins in the 1960s and is a common time series prediction model [38]. The ARIMA model addresses non-stationarity and seasonality by capturing the time series trend in the sequence data, as well as controlling the autocorrelation of the sequence, which enables the identification of periodicity and long-term trends in the data [43]. Model fitness was assessed by testing for autocorrelation remaining in the residuals using the Ljung-Box Pierce test.

## 3. Results

Our dataset consisted of 11,105,974 food purchase transactions. Table 1 provides a detailed summary of food purchases and nutrient composition across three time periods: pre-COVID, COVID, and post-COVID pandemic restrictions. 24.9% of purchases occurred before COVID, 25.6% during the pandemic, and 49.5% after restrictions were lifted. Ultra-processed foods consistently accounted for the highest purchases (76–77%) across all periods, while processed culinary ingredients remained stable at around 1.5% accounting for the least purchases.

## 3.1 Effect of COVID-19 restrictions on food purchases

We estimated the impact of the COVID-19 pandemic on food purchases by calculating the weekly change in proportions of NOVA classification, as summarised in Table 2 and Fig 1. Table 3, Figs 2–4 summarise the impact of the COVID-19 pandemic on nutrient composition per 100g/100ml of purchased food by calculating the weekly change in mean values of proximates, minerals, and vitamins.

### 3.1.1 Before COVID-19.

**NOVA:** During the pre-COVID period, weekly purchases of processed culinary ingredients ($\beta_1$=-0.0045%) and unprocessed/minimally processed foods ($\beta_1$=-0.0295%) showed a declining trend. In contrast, purchases of processed foods ($\beta_1$=0.0035%) and ultra-processed foods ($\beta_1$=0.0322%) demonstrated increasing trends over the same period (Table 2; Fig 1).

**Proximates:** Pre-COVID weekly trends of energy ($\beta_1$=0.6648 kcal) and carbohydrates ($\beta_1$=0.0103g) showed increasing patterns. In contrast, proteins exhibited a weekly declining trend of ($\beta_1$=-0.0037g) (Table 3; Fig 2).

**Minerals:** Pre-COVID trends of calcium ($\beta_1$=-0.0264 mg), iron ($\beta_1$=-0.0019 mg), magnesium ($\beta_1$=-0.0203 mg), potassium ($\beta_1$=-0.1702 mg), sodium ($\beta_1$=-0.5827 mg), and selenium ($\beta_1$=-0.0044 µg) exhibited a weekly decrease (Table 3; Fig 3).

**Vitamins:** Pre-COVID weekly trend of vitamin C increased by ($\beta_1$=0.0060mg). In contrast, purchased vitamin A, thiamin, niacin, dietary folate equivalent, and vitamin B12 exhibited a weekly significant decline of ($\beta_1$=-0.4778mcg), ($\beta_1$=-0.0002mg), ($\beta_1$=-0.0007mg), ($\beta_1$=-0.0217mcg), and ($\beta_1$=-0.0003mcg) respectively (Table 3; Fig 4).

### 3.1.2 During COVID-19.

**NOVA:** The start of lockdown was associated with an increase in purchases of processed foods ($\beta_2$=0.4071%), and a sharp decline in purchases of ultra-processed foods ($\beta_2$=-2.4718%). Unprocessed/minimally processed foods also increased at lockdown onset ($\beta_2$=1.6701%). Across all NOVA categories, no sustained long-term changes ($\beta_3$) were observed. (Table 2; Fig 1)

**Proximates:** The start of lockdown was associated with a significant increase in purchased fibre ($\beta_2$=0.3828g). The COVID period was significantly associated with a weekly decline in purchased carbohydrates ($\beta_3$=-0.0144g). Consistent significant increase in the COVID trends of purchased proteins by ($\beta_2$=0.2738g) in the short-term and by ($\beta_3$=0.0027g) in the long-term (Table 3; Fig 2).

**Minerals:** Lockdown onset was associated with a sharp increase in iron ($\beta_2$=0.1427mg), magnesium ($\beta_2$=2.9282mg), phosphorus ($\beta_2$=11.1537mg), potassium ($\beta_2$=28.3364mg), sodium ($\beta_2$=55.7834mg) and zinc ($\beta_2$=0.0373mg). While there were short-term increases in purchased iron (mg), magnesium (mg), phosphorus (mg), potassium (mg), sodium (mg) and zinc (mg) following the lockdown onset, these did not translate into a significant sustained effect ($\beta_3$). There was a consistent increase in the COVID trend of selenium by ($\beta_2$=0.1995mcg) in the short-term and by ($\beta_3$=0.0057g) in the long-term. The pandemic trend of calcium showed variation; increased sharply by ($\beta_2$=11.4832mg) in the short-term, with a long-term weekly decrease by ($\beta_3$=-0.1025mg) (Table 3; Fig 3).

**Vitamins:** The start of lockdown was associated with a sharp increase in dietary folate ($\beta_2$=2.3969mcg) and a decrease in riboflavin ($\beta_2$=-0.1619mg). Although immediate changes were observed in riboflavin and dietary folate, these effects were not sustained over time ($\beta_3$). While there was no immediate change in trend ($\beta_2$) on thiamin and vitamin C, the COVID period was associated with a weekly increase in thiamin ($\beta_3$=0.0002mg) and a decline in vitamin C ($\beta_3$=-0.0117mg). Consistent increases were observed in both the short-term ($\beta_2$) and long-term ($\beta_3$) trends for vitamin A-RE ($\beta_2$=18.3207 mcg; $\beta_3$=0.3077 mcg), niacin ($\beta_2$=0.0684 mg; $\beta_3$=0.0015 mg), and vitamin B12 ($\beta_2$=0.0309 mcg; $\beta_3$=0.0003 mcg) (Table 3; Fig 4).

### 3.1.3 After COVID-19.

**NOVA:** Following the end of COVID restrictions, purchase trends of processed culinary ingredients, ultra-processed foods, and unprocessed/minimally processed foods showed no immediate ($\beta_4$) and sustained effects ($\beta_5$). In contrast, purchases of processed foods exhibited a short-term increase ($\beta_4$=0.3433%), followed by a delayed long-term weekly decline ($\beta_5$=-0.0130%). (Table 2; Fig 1).

**Proximates:** At lockdown end, no immediate short-term changes were observed across all the trends of proximates ($\beta_4$). Post-COVID trends of water (g), protein (g), fibre (g) and cholesterol (mg) exhibited no sustained changes ($\beta_5$).

PLOS Global Public Health

**Table 1. Descriptive characteristics of the transaction data from 2018-2023.**

| Variable | Category | Summary statistic | Overall N=11105974 | Period* | | |
|---|---|---|---|---|---|---|
| | | | | Pre-Covid N=2763060 | Covid N=2844357 | Post-Covid N=5498557 |
| NOVA food classification | Processed Culinary Ingredients | n (%) | 168214 (1.51) | 42818 (1.55) | 43534 (1.53) | 81862 (1.49) |
| | Processed foods | n (%) | 310801 (2.80) | 59147 (2.14) | 86116 (3.03) | 165538 (3.01) |
| | Ultra-processed foods | n (%) | 8511558 (76.64) | 2115260 (76.55) | 2157120 (75.84) | 4239178 (77.10) |
| | Unprocessed/ Minimally processed foods | n (%) | 2115401 (19.05) | 545835 (19.75) | 557587 (19.60) | 1011979 (18.40) |
| Proximates | Energy (kcal) | Mean (SD) | 623.97 (662.76) | 573.74 (624.01) | 638.65 (676.41) | 641.76 (673.27) |
| | | Median (Q1, Q3) | 306.00 (249.00, 463.00) | 306.00 (249.00, 463.00) | 306.00 (245.00, 537.00) | 306.00 (249.00, 541.00) |
| | | Min, Max | 2.00, 3692.00 | 2.00, 3082.00 | 2.00, 3692.00 | 2.00, 3692.00 |
| | Water (g) | Mean (SD) | 37.67 (31.91) | 38.50 (31.46) | 37.41 (32.06) | 37.38 (32.06) |
| | | Median (Q1, Q3) | 29.00 (7.90, 74.60) | 29.00 (12.20, 75.40) | 29.00 (6.10, 71.80) | 29.00 (6.10, 71.80) |
| | | Min, Max | 0.10, 99.95 | 0.20, 99.95 | 0.10, 99.95 | 0.10, 99.95 |
| | Protein (g) | Mean (SD) | 6.07 (4.30) | 6.01 (3.98) | 6.05 (4.29) | 6.11 (4.45) |
| | | Median (Q1, Q3) | 6.60 (3.90, 7.00) | 6.60 (3.90, 7.00) | 6.60 (3.90, 7.00) | 6.60 (3.90, 7.00) |
| | | Min, Max | 0.10, 84.40 | 0.10, 84.40 | 0.10, 84.40 | 0.10, 84.40 |
| | Fat (g) | Mean (SD) | 12.06 (13.11) | 11.45 (12.76) | 12.31 (13.09) | 12.25 (13.29) |
| | | Median (Q1, Q3) | 6.80 (4.20, 15.30) | 6.80 (4.20, 15.00) | 6.80 (4.20, 15.30) | 6.80 (4.20, 15.30) |
| | | Min, Max | 0.10, 100.00 | 0.10, 100.00 | 0.10, 100.00 | 0.10, 100.00 |
| | Carbohydrate available (g) | Mean (SD) | 44.49 (24.80) | 44.15 (24.36) | 44.16 (24.93) | 44.83 (24.96) |
| | | Median (Q1, Q3) | 52.80 (17.70, 59.40) | 52.80 (17.70, 58.80) | 52.80 (17.70, 60.00) | 52.80 (17.70, 60.00) |
| | | Min, Max | 0.10, 102.70 | 0.10, 101.30 | 0.10, 101.30 | 0.10, 102.70 |
| | Fibre (g) | Mean (SD) | 4.02 (5.16) | 3.87 (4.82) | 4.17 (5.50) | 4.01 (5.16) |
| | | Median (Q1, Q3) | 2.90 (2.30, 3.10) | 2.90 (2.90, 3.00) | 2.90 (2.30, 3.10) | 2.90 (2.30, 3.10) |
| | | Min, Max | 0.10, 52.30 | 0.20, 52.30 | 0.20, 52.30 | 0.10, 52.30 |
| | Cholesterol (mg) | Mean (SD) | 22.58 (50.19) | 21.36 (49.40) | 22.42 (46.44) | 23.34 (52.48) |
| | | Median (Q1, Q3) | 3.40 (1.50, 32.00) | 3.40 (1.50, 32.00) | 3.40 (1.50, 34.00) | 3.40 (1.50, 32.00) |
| | | Min, Max | 0.10, 418.00 | 0.10, 418.00 | 0.10, 418.00 | 0.10, 418.00 |
| Minerals | Calcium (mg) | Mean (SD) | 95.82 (181.90) | 94.78 (165.82) | 100.13 (199.21) | 94.11 (180.30) |
| | | Median (Q1, Q3) | 90.00 (37.00, 110.00) | 90.00 (42.00, 110.00) | 90.00 (37.00, 110.00) | 90.00 (36.00, 110.00) |
| | | Min, Max | 1.00, 4280.00 | 1.00, 4280.00 | 1.00, 4280.00 | 1.00, 4280.00 |
| | Iron (mg) | Mean (SD) | 2.11 (4.34) | 2.09 (4.03) | 2.11 (4.28) | 2.12 (4.53) |
| | | Median (Q1, Q3) | 1.70 (0.20, 2.50) | 1.80 (0.20, 2.50) | 1.80 (0.20, 2.50) | 1.70 (0.20, 2.50) |
| | | Min, Max | 0.01, 123.60 | 0.01, 123.60 | 0.01, 123.60 | 0.01, 123.60 |
| | Magnesium (mg) | Mean (SD) | 34.53 (51.93) | 33.41 (48.83) | 35.06 (52.80) | 34.84 (52.98) |
| | | Median (Q1, Q3) | 27.00 (13.00, 27.00) | 27.00 (13.00, 27.00) | 27.00 (13.00, 30.00) | 27.00 (13.00, 29.00) |
| | | Min, Max | 1.00, 430.00 | 1.00, 420.00 | 1.00, 430.00 | 1.00, 430.00 |
| | Phosphorus (mg) | Mean (SD) | 139.33 (315.23) | 134.96 (286.19) | 143.50 (347.83) | 139.39 (311.43) |
| | | Median (Q1, Q3) | 100.00 (96.00, 143.00) | 100.00 (98.00, 143.00) | 100.00 (96.00, 143.00) | 100.00 (96.00, 143.00) |
| | | Min, Max | 1.00, 8410.00 | 1.00, 8410.00 | 1.00, 8410.00 | 1.00, 8410.00 |
| | Potassium (mg) | Mean (SD) | 299.22 (617.23) | 294.78 (602.40) | 311.07 (639.48) | 295.35 (612.87) |
| | | Median (Q1, Q3) | 210.00 (126.00, 253.00) | 210.00 (140.00, 245.00) | 210.00 (129.00, 270.00) | 191.00 (120.00, 245.00) |
| | | Min, Max | 1.00, 10200.00 | 1.00, 10200.00 | 1.00, 10200.00 | 1.00, 10200.00 |

*(Continued)*

**Table 1.** (Continued)

| Variable | Category | Summary statistic | Overall N = 11105974 | Period* | | |
| --- | --- | --- | --- | --- | --- | --- |
| | | | | Pre-Covid N = 2763060 | Covid N = 2844357 | Post-Covid N = 5498557 |
| | Sodium (mg) | Mean (SD) | 287.99 (1717.08) | 297.95 (1886.52) | 289.52 (1712.71) | 282.12 (1626.44) |
| | | Median (Q1, Q3) | 100.00 (45.00, 170.00) | 100.00 (45.00, 155.00) | 100.00 (45.00, 170.00) | 100.00 (45.00, 170.00) |
| | | Min, Max | 1.00, 38500.00 | 1.00, 38500.00 | 1.00, 38500.00 | 1.00, 38500.00 |
| | Zinc (mg) | Mean (SD) | 0.78 (0.67) | 0.75 (0.61) | 0.78 (0.69) | 0.79 (0.70) |
| | | Median (Q1, Q3) | 0.60 (0.50, 0.80) | 0.60 (0.50, 0.74) | 0.60 (0.50, 0.80) | 0.60 (0.50, 0.80) |
| | | Min, Max | 0.01, 8.00 | 0.01, 8.00 | 0.01, 8.00 | 0.01, 8.00 |
| | Selenium (mcg) | Mean (SD) | 6.53 (7.88) | 6.34 (6.84) | 6.37 (7.82) | 6.71 (8.41) |
| | | Median (Q1, Q3) | 4.00 (2.00, 9.00) | 4.00 (2.00, 9.00) | 4.00 (2.00, 9.00) | 5.00 (2.00, 9.00) |
| | | Min, Max | 0.20, 254.00 | 0.20, 79.00 | 0.20, 254.00 | 0.20, 254.00 |
| Vitamins | Vitamin A-RE (mcg) | Mean (SD) | 123.72 (645.60) | 136.52 (752.84) | 124.01 (642.82) | 117.13 (585.82) |
| | | Median (Q1, Q3) | 47.00 (34.00, 101.00) | 47.00 (34.00, 101.00) | 47.00 (34.00, 101.00) | 47.00 (29.00, 101.00) |
| | | Min, Max | 1.00, 11000.00 | 1.00, 11000.00 | 1.00, 11000.00 | 1.00, 11000.00 |
| | Thiamin (mg) | Mean (SD) | 0.17 (0.19) | 0.17 (0.16) | 0.17 (0.17) | 0.17 (0.20) |
| | | Median (Q1, Q3) | 0.16 (0.09, 0.20) | 0.19 (0.09, 0.20) | 0.16 (0.09, 0.20) | 0.16 (0.09, 0.20) |
| | | Min, Max | 0.01, 3.78 | 0.01, 2.80 | 0.01, 3.78 | 0.01, 3.78 |
| | Riboflavin (mg) | Mean (SD) | 0.25 (4.71) | 0.32 (6.70) | 0.20 (2.81) | 0.23 (4.22) |
| | | Median (Q1, Q3) | 0.10 (0.03, 0.20) | 0.10 (0.03, 0.20) | 0.12 (0.03, 0.20) | 0.10 (0.03, 0.20) |
| | | Min, Max | 0.01, 272.00 | 0.01, 272.00 | 0.01, 272.00 | 0.01, 272.00 |
| | Niacin (mg) | Mean (SD) | 2.25 (4.70) | 2.17 (4.38) | 2.23 (4.67) | 2.31 (4.87) |
| | | Median (Q1, Q3) | 1.30 (0.40, 2.10) | 2.00 (0.40, 2.10) | 1.10 (0.40, 2.10) | 1.10 (0.40, 2.10) |
| | | Min, Max | 0.05, 42.90 | 0.10, 42.90 | 0.05, 42.90 | 0.05, 42.90 |
| | Dietary Folate Equivalent (mcg) | Mean (SD) | 23.58 (96.78) | 22.99 (93.09) | 24.11 (106.00) | 23.61 (93.58) |
| | | Median (Q1, Q3) | 12.00 (9.00, 17.00) | 12.00 (10.00, 16.00) | 12.00 (9.00, 17.00) | 12.00 (9.00, 17.00) |
| | | Min, Max | 0.40, 4000.00 | 1.00, 4000.00 | 0.40, 4000.00 | 0.40, 4000.00 |
| | Vitamin B12 (mcg) | Mean (SD) | 0.54 (0.65) | 0.54 (0.61) | 0.55 (0.64) | 0.54 (0.68) |
| | | Median (Q1, Q3) | 0.30 (0.15, 1.00) | 0.30 (0.15, 1.00) | 0.30 (0.15, 1.00) | 0.30 (0.15, 1.00) |
| | | Min, Max | 0.05, 12.00 | 0.05, 5.00 | 0.05, 5.00 | 0.05, 12.00 |
| | Vitamin C (mg) | Mean (SD) | 6.93 (11.85) | 7.07 (14.89) | 7.32 (12.00) | 6.66 (9.93) |
| | | Median (Q1, Q3) | 3.00 (1.00, 14.00) | 2.00 (1.00, 14.00) | 3.00 (1.00, 14.00) | 3.00 (1.00, 14.00) |
| | | Min, Max | 0.10, 295.00 | 0.15, 295.00 | 0.10, 295.00 | 0.10, 295.00 |

Note: Mixed Dishes and Fast Foods/Starchy Roots and Tubers Transactions omitted in ITS analysis as data points limited in duration and coverage; * = column percentages

However, there was a long-term post-COVID weekly increase in carbohydrates ($\beta_5 = 0.0182$g). In contrast, post-COVID trends indicated a weekly decline of energy ($\beta_5 = -0.8287$g) and a delayed weekly decline of fat ($\beta_5 = -0.0199$g) (Table 3; Fig 2).

**Minerals:** After the pandemic, no short-term ($\beta_4$) and long-term ($\beta_5$) trend changes were observed in iron (mg), magnesium (mg), phosphorus (mg), potassium (mg), and zinc (mg). The end of lockdown was associated with an increase

**Table 2. Parameter estimates, confidence intervals and Z-test p-values from the full ITS-GLS models predicting the weekly proportion of NOVA classification groups.**

| Variable | Category | Optimal ITS-Generalised Least Squares model | Intercept ($\beta_0$) | | Pre-COVID ($\beta_1$) | | Start of Lockdown ($\beta_2$) | | COVID Period ($\beta_3$) | | End of Lockdown ($\beta_4$) | | Post-COVID ($\beta_5$) | |
|---|---|---|---|---|---|---|---|---|---|---|---|---|---|---|
| | | | Coefficient (95% CI) | Z test p-value | Coefficient (95% CI) | Z test p-value | Coefficient (95% CI) | Z test p-value | Coefficient (95% CI) | Z test P-value | Coefficient (95% CI) | Z test p-value | Coefficient (95% CI) | Z test p-value |
| NOVA food classification | Processed Culinary Ingredients | corAR-MA(p=3, q=3) | 1.8766 (1.6777, 2.0754) | **<0.001** | -0.0045 (-0.0073, -0.0017) | **0.002** | 0.1905 (-0.0128, 0.3939) | 0.066 | 0.0037 (-0.0018, 0.0093) | 0.184 | 0.1253 (-0.0773, 0.3279) | 0.225 | -0.0015 (-0.0070, 0.0040) | 0.596 |
| | Processed foods | corAR-MA(p=2, q=2) | 1.9619 (1.8315, 2.0924) | **<0.001** | 0.0035 (0.0014, 0.0056) | **0.001** | 0.4071 (0.1483, 0.6659) | **0.002** | 0.0016 (-0.0023, 0.0055) | 0.422 | 0.3433 (0.0862, 0.6003) | **0.009** | -0.0138 (-0.0177, -0.0099) | **<0.001** |
| | Ultra-processed foods | corAR-MA(p=0, q=3) | 74.2951 (72.9151, 75.6751) | **<0.001** | 0.0322 (0.0121, 0.0522) | **0.002** | -2.4718 (-4.2835, -0.6601) | **0.007** | -0.0241 (-0.0629, 0.0148) | 0.224 | -0.2200 (-2.0235, 1.5835) | 0.811 | 0.0103 (-0.0285, 0.0491) | 0.603 |
| | Unprocessed/ Minimally processed foods | corAR-MA(p=0, q=3) | 21.7773 (20.5812, 22.9734) | **<0.001** | -0.0295 (-0.0468, -0.0121) | **0.001** | 1.6701 (0.1177, 3.2224) | **0.035** | 0.0180 (-0.0155, 0.0516) | 0.292 | -0.1605 (-1.7059, 1.3849) | 0.839 | 0.0030 (-0.0306, 0.0365) | 0.863 |

Note: corARMA = accounts for autocorrelation; Mixed Dishes and Fast Foods/Starchy Roots and Tubers Transactions omitted in ITS analysis as data points limited in duration and coverage

in selenium ($\beta_4$ = 0.2112mcg), while post-COVID trends showed a weekly increase in sodium ($\beta_5$ = 0.8018mg). Calcium showed a consistent upward trend, with a short-term increase ($\beta_4$ = 2.4178mg) followed by a sustained long-term rise ($\beta_5$ = 0.0804mg) (Table 3; Fig 3).

**Vitamins:** Post-pandemic, no short-term ($\beta_4$) and long-term ($\beta_5$) trend changes were observed in vitamin A-RE (mcg), thiamin (mg), riboflavin (mg), niacin (mg), dietary folate equivalent (mcg) and vitamin C (mg). The post-COVID trend of vitamin B12 exhibited a short term increase ($\beta_4$ = 0.0191mcg) followed by a delayed long-term decline ($\beta_5$ = -0.0007mcg) (Table 3; Fig 4).

### 3.2 Sensitivity analyses

Table 4 summarises model fit diagnostics and accuracy performance metrics comparing the full ITS-GLS and ARIMA models.

GLS models generally had lower AIC and BIC values across nearly all categories. Overall, the GLS model is superior in capturing the dynamics and structural changes and fits the training data better compared to the ARIMA model. Thus, the model fit results of AIC/BIC support our modelling choice of using GLS. However, ARIMA models consistently showed lower prediction error on test data (e.g., lower RMSE and MAE), suggesting improved forecasting accuracy, which indicates better generalisation to unseen data.

Fig 5 summarises the results of the sensitivity analysis for each coefficient ($\beta_0$, $\beta_1$, $\beta_2$, $\beta_3$, $\beta_4$, $\beta_5$), by checking whether significance and direction (positive/negative) are robust across GLS and ARIMA models. Using ARIMA models yielded results that were largely consistent with those from the main GLS analysis for majority (87.8%) of the coefficients, with the exception of 1) stable coefficient signs significant in GLS models only (10.3%); 2) stable coefficient signs significant in ARIMA models only (0.6%); and 3) non-significant different coefficient signs (1.3%). These findings suggest that the estimated coefficients are generally robust in both direction and significance across modeling approaches, reinforcing the reliability of the main GLS results.

PLOS Global Public Health

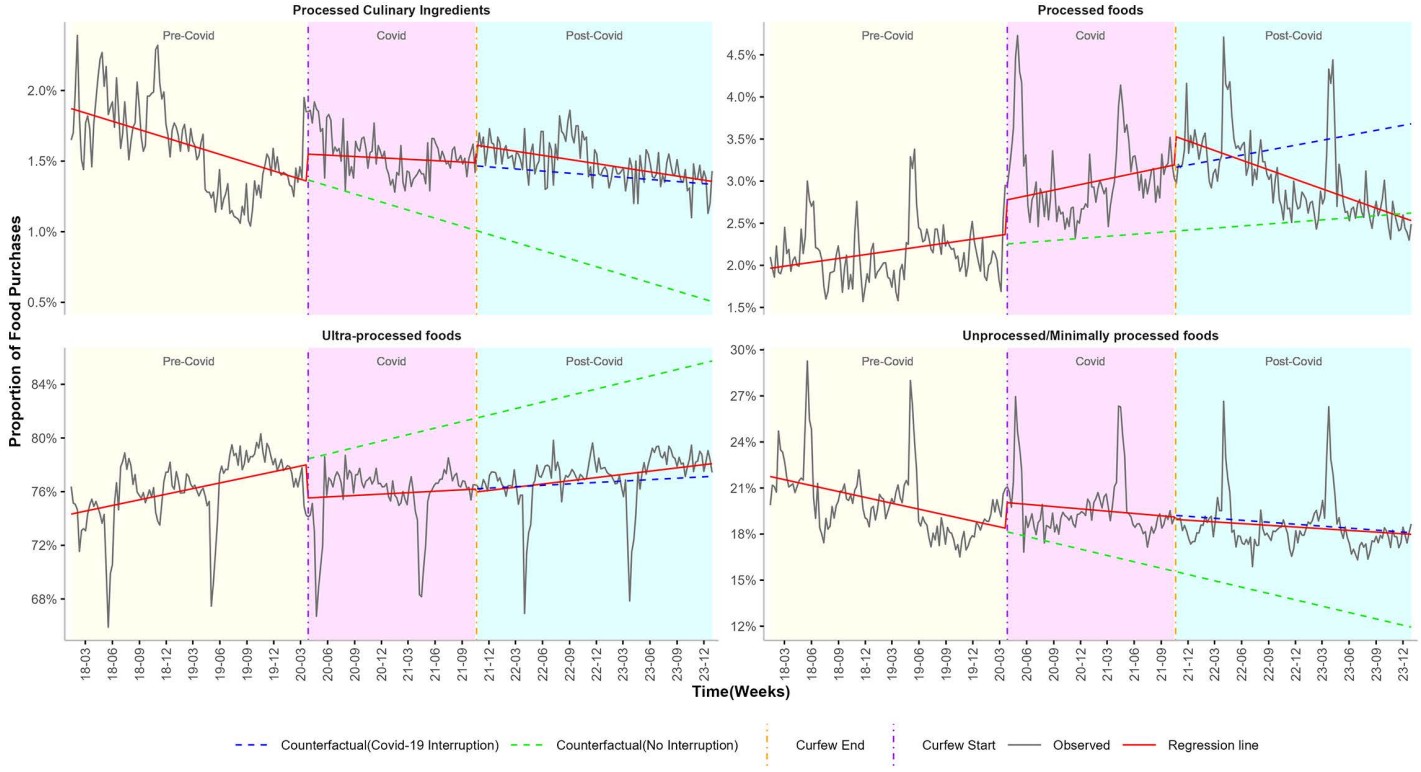

**Fig 1. ITS-GLS model weekly proportion estimates and the counterfactual of NOVA classification before, during and after pandemic restrictions.** Curfew start vertical line = 27th March 2020 (start of pandemic restrictions). Curfew end vertical line = 20th October 2021 (end of pandemic restrictions). Predictions were estimated in two parts: by extrapolating the pre-pandemic trend and by extrapolating the combined pre-pandemic and pandemic trend.

## 4. Discussion

This paper, using large-scale consumer purchase data with an interrupted time series design, presents evidence of significant changes in food purchase patterns before, during, and after the COVID-19 pandemic, with implications for population diet quality and nutrition-related health risks. While the interrupted time series analysis identified statistically significant level and trend changes in the outcomes following the COVID-19 interventions, it also identified substantive magnitude. Statistical significance reflects the likelihood that the observed change is unlikely to have occurred by chance, given the model assumptions and sample size. In contrast, substantive magnitude refers to the magnitude of the effect and whether it represents a positive or negative improvement in the outcomes. In our context, both the statistical significance and the estimated magnitude of change guided interpretation and conclusions about the impact of the interventions.

Our findings during the pre-COVID period reveal that trends were largely consistent with nutrition transition theories [44–46]. The onset of the COVID-19 pandemic was associated with immediate transient disruptions in food purchasing behavior. These changes reflect the global population's rapid adaptation to pandemic-related restrictions, perceived food security threats, and shifting lifestyle priorities during lockdown [47,48]. While some purchase trends suggest compensatory improvements in dietary quality, others point to potential nutritional vulnerabilities and volatility in consumer behavior during public health crises.

Table 3. Parameter estimates, confidence intervals and Z-test p-values from the full ITS-GLS models predicting the weekly mean nutrient values per 100g/100ml of food.

| Variable | Category | Optimal ITS -Generalised Least Squares model | Intercept ($\beta_0$) Coefficient (95% CI) | Z test p-value | Pre-COVID ($\beta_1$) Coefficient (95% CI) | Z test p-value | Start of Lockdown ($\beta_2$) Coefficient (95% CI) | Z test p-value | COVID Period ($\beta_3$) Coefficient (95% CI) | Z test P-value | End of Lockdown ($\beta_4$) Coefficient (95% CI) | Z test p-value | Post-COVID ($\beta_5$) Coefficient (95% CI) | Z test p-value |
|---|---|---|---|---|---|---|---|---|---|---|---|---|---|---|
| Proxi-mates | Energy (kcal) | corARMA (p=1, q=2) | 532.2056 (503.1463, 561.2650) | <0.001 | 0.6648 (0.2499, 1.0797) | 0.002 | -0.1722 (-34.1237, 33.7794) | 0.992 | -0.0314 (-0.8450, 0.7821) | 0.940 | -8.1101 (-41.9199, 25.6997) | 0.638 | -0.8247 (-1.6382, -0.0112) | 0.047 |
| | Water (g) | corARMA (p=2, q=4) | 38.9262 (37.4737, 40.3787) | <0.001 | -0.0068 (-0.0273, 0.0137) | 0.518 | -0.9686 (-2.5368, 0.5996) | 0.226 | 0.0112 (-0.0290, 0.0514) | 0.584 | 0.2845 (-1.2775, 1.8465) | 0.721 | -0.0105 (-0.0506, 0.0297) | 0.609 |
| | Protein (g) | corARMA (p=1, q=0) | 6.2559 (6.1867, 6.3251) | <0.001 | -0.0037 (-0.0047, -0.0027) | <0.001 | 0.2738 (0.1744, 0.3732) | <0.001 | 0.0027 (0.0007, 0.0047) | 0.008 | 0.0677 (-0.0312, 0.1667) | 0.180 | 0.0015 (-0.0005, 0.0035) | 0.135 |
| | Fat (g) | corARMA (p=4, q=3) | 11.4199 (11.0338, 11.8060) | <0.001 | 0.0009 (-0.0046, 0.0064) | 0.756 | 0.2956 (-0.1551, 0.7463) | 0.199 | 0.0102 (-0.0006, 0.0210) | 0.065 | 0.0154 (-0.4335, 0.4643) | 0.946 | -0.0199 (-0.0307, -0.0091) | <0.001 |
| | Carbo-hydrate available (g) | corARMA (p=2, q=1) | 43.3490 (42.8863, 43.8116) | <0.001 | 0.0103 (0.0028, 0.0177) | 0.007 | -0.0888 (-0.9822, 0.8045) | 0.845 | -0.0144 (-0.0281, -0.0007) | 0.040 | -0.1657 (-1.0533, 0.7219) | 0.714 | 0.0182 (0.0044, 0.0319) | 0.009 |
| | Fibre (g) | corARMA (p=3, q=3) | 4.0776 (3.8519, 4.3032) | <0.001 | -0.0026 (-0.0058, 0.0007) | 0.119 | 0.3828 (0.1101, 0.6555) | 0.006 | 0.0026 (-0.0036, 0.0089) | 0.413 | -0.0717 (-0.3434, 0.2000) | 0.605 | -0.0013 (-0.0076, 0.0049) | 0.674 |
| | Cho-lesterol (mg) | corARMA (p=3, q=3) | 20.4396 (18.8550, 22.0242) | <0.001 | 0.0137 (-0.0084, 0.0359) | 0.224 | 1.2700 (-0.3127, 2.8527) | 0.116 | -0.0216 (-0.0653, 0.0220) | 0.332 | 0.9792 (-0.5969, 2.5553) | 0.223 | -0.0029 (-0.0465, 0.0408) | 0.898 |
| Minerals | Calcium (mg) | corARMA (p=4, q=1) | 96.9558 (95.7809, 98.1307) | <0.001 | -0.0264 (-0.0454, -0.0074) | 0.006 | 11.4832 (9.0983, 13.8682) | <0.001 | -0.1025 (-0.1368, -0.0682) | <0.001 | 2.4178 (0.0495, 4.7862) | 0.045 | 0.0804 (0.0461, 0.1147) | <0.001 |
| | Iron (mg) | corARMA (p=3, q=2) | 2.2153 (2.1339, 2.2967) | <0.001 | -0.0019 (-0.0031, -0.0007) | 0.002 | 0.1427 (0.0258, 0.2595) | 0.017 | 0.0012 (-0.0011, 0.0036) | 0.301 | 0.0319 (-0.0844, 0.1482) | 0.591 | 0.0006 (-0.0017, 0.0030) | 0.598 |
| | Mag-nesium (mg) | corARMA (p=3, q=2) | 34.8815 (33.8737, 35.8892) | <0.001 | -0.0203 (-0.0351, -0.0055) | 0.007 | 2.9282 (1.5134, 4.3430) | <0.001 | 0.0098 (-0.0190, 0.0387) | 0.504 | 0.5399 (-0.8680, 1.9478) | 0.452 | 0.0045 (-0.0244, 0.0333) | 0.762 |
| | Phos-phorus (mg) | corARMA (p=0, q=4) | 135.7179 (130.8101, 140.6256) | <0.001 | -0.0004 (-0.0723, 0.0716) | 0.992 | 11.1537 (4.3685, 17.9388) | 0.001 | -0.0785 (-0.2184, 0.0614) | 0.271 | 2.8781 (-3.8749, 9.6312) | 0.404 | 0.0087 (-0.1312, 0.1486) | 0.903 |
| | Potas-sium (mg) | corARMA (p=3, q=2) | 306.8534 (298.0547, 315.6521) | <0.001 | -0.1702 (-0.2994, -0.0410) | 0.010 | 28.3364 (16.0789, 40.5940) | <0.001 | 0.0512 (-0.2005, 0.3029) | 0.690 | 0.4704 (-11.7287, 12.6694) | 0.940 | -0.0723 (-0.3239, 0.1794) | 0.574 |
| | Sodium (mg) | corARMA (p=4, q=3) | 338.8413 (316.5386, 361.1441) | <0.001 | -0.5827 (-0.9077, -0.2578) | <0.001 | 55.7834 (26.2428, 85.3240) | <0.001 | -0.2257 (-0.8552, 0.4039) | 0.482 | 19.0765 (-10.3370, 48.4899) | 0.204 | 0.8018 (0.1722, 1.4313) | 0.013 |
| | Zinc (mg) | corARMA (p=3, q=2) | 0.7533 (0.7352, 0.7714) | <0.001 | -0.0001 (-0.0003, 0.0002) | 0.570 | 0.0373 (0.0130, 0.0616) | 0.003 | 0.0001 (-0.0004, 0.0006) | 0.681 | 0.0106 (-0.0136, 0.0347) | 0.392 | -0.0001 (-0.0006, 0.0004) | 0.712 |

*(Continued)*

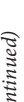

Table 3. (Continued)

| Variable | Category | Optimal ITS-Generalised Least Squares model | Intercept ($\beta_0$) Coefficient (95% CI) | Z test p-value | Pre-COVID ($\beta_1$) Coefficient (95% CI) | Z test p-value | Start of Lockdown ($\beta_2$) Coefficient (95% CI) | Z test p-value | COVID Period ($\beta_3$) Coefficient (95% CI) | Z test P-value | End of Lockdown ($\beta_4$) Coefficient (95% CI) | Z test p-value | Post-COVID ($\beta_5$) Coefficient (95% CI) | Z test p-value |
|---|---|---|---|---|---|---|---|---|---|---|---|---|---|---|
| | | Selenium (mcg) corARMA(p=4, q=4) | 6.6205 (6.5188, 6.7222) | <0.001 | -0.0044 (-0.0059, -0.0029) | <0.001 | 0.1995 (0.0524, 0.3466) | 0.008 | 0.0057 (0.0028, 0.0086) | <0.001 | 0.2112 (0.0648, 0.3576) | 0.005 | 0.0000 (-0.0030, 0.0029) | 0.978 |
| | Vitamins | Vitamin A-RE (mcg) corARMA(p=3, q=2) | 167.3124 (159.9337, 174.6912) | <0.001 | -0.4778 (-0.5873, -0.3684) | <0.001 | 18.3207 (7.4157, 29.2258) | 0.001 | 0.3077 (0.0947, 0.5207) | 0.005 | 3.1252 (-7.7245, 13.9749) | 0.572 | 0.1264 (-0.0866, 0.3394) | 0.245 |
| | | Thiamin (mg) corARMA(p=1, q=0) | 0.1827 (0.1801, 0.1853) | <0.001 | -0.0002 (-0.0002, -0.0001) | <0.001 | 0.0035 (-0.0001, 0.0072) | 0.060 | 0.0002 (0.0001, 0.0002) | <0.001 | 0.0016 (-0.0021, 0.0052) | 0.396 | 0.0001 (0.0000, 0.0001) | 0.143 |
| | | Riboflavin (mg) corARMA(p=4, q=4) | 0.2948 (0.2346, 0.3550) | <0.001 | 0.0004 (-0.0005, 0.0012) | 0.422 | -0.1619 (-0.2407, -0.0830) | <0.001 | 0.0003 (-0.0014, 0.0020) | 0.745 | -0.0126 (-0.0911, 0.0659) | 0.753 | -0.0003 (-0.0020, 0.0014) | 0.726 |
| | | Niacin (mg) corARMA(p=2, q=1) | 2.2130 (2.1697, 2.2563) | <0.001 | -0.0007 (-0.0014, -0.0001) | 0.022 | 0.0684 (0.0050, 0.1319) | 0.034 | 0.0015 (0.0003, 0.0028) | 0.016 | 0.0150 (-0.0481, 0.0781) | 0.641 | -0.0002 (-0.0014, 0.0010) | 0.751 |
| | | Dietary Folate Equivalent (mcg) corARMA(p=0, q=4) | 24.4765 (23.2392, 25.7138) | <0.001 | -0.0217 (-0.0400, -0.0034) | 0.020 | 2.3969 (0.6145, 4.1793) | 0.008 | 0.0150 (-0.0205, 0.0506) | 0.408 | 1.1585 (-0.6152, 2.9322) | 0.200 | -0.0167 (-0.0523, 0.0188) | 0.356 |
| | | Vitamin B12 (mcg) corARMA(p=4, q=3) | 0.5520 (0.5484, 0.5555) | <0.001 | -0.0003 (-0.0003, -0.0002) | <0.001 | 0.0309 (0.0240, 0.0379) | <0.001 | 0.0003 (0.0002, 0.0004) | <0.001 | 0.0191 (0.0122, 0.0260) | <0.001 | -0.0007 (-0.0007, -0.0006) | <0.001 |
| | | Vitamin C (mg) corARMA(p=3, q=3) | 6.6775 (6.4236, 6.9313) | <0.001 | 0.0060 (0.0023, 0.0098) | 0.002 | 0.2193 (-0.1322, 0.5708) | 0.221 | -0.0117 (-0.0190, -0.0045) | 0.002 | 0.1317 (-0.2182, 0.4815) | 0.461 | -0.0046 (-0.0118, 0.0027) | 0.217 |

Note: corARMA = accounts for autocorrelation; Mixed Dishes and Fast Foods/Starchy Roots and Tubers Transactions omitted in ITS analysis as data points limited in duration and coverage

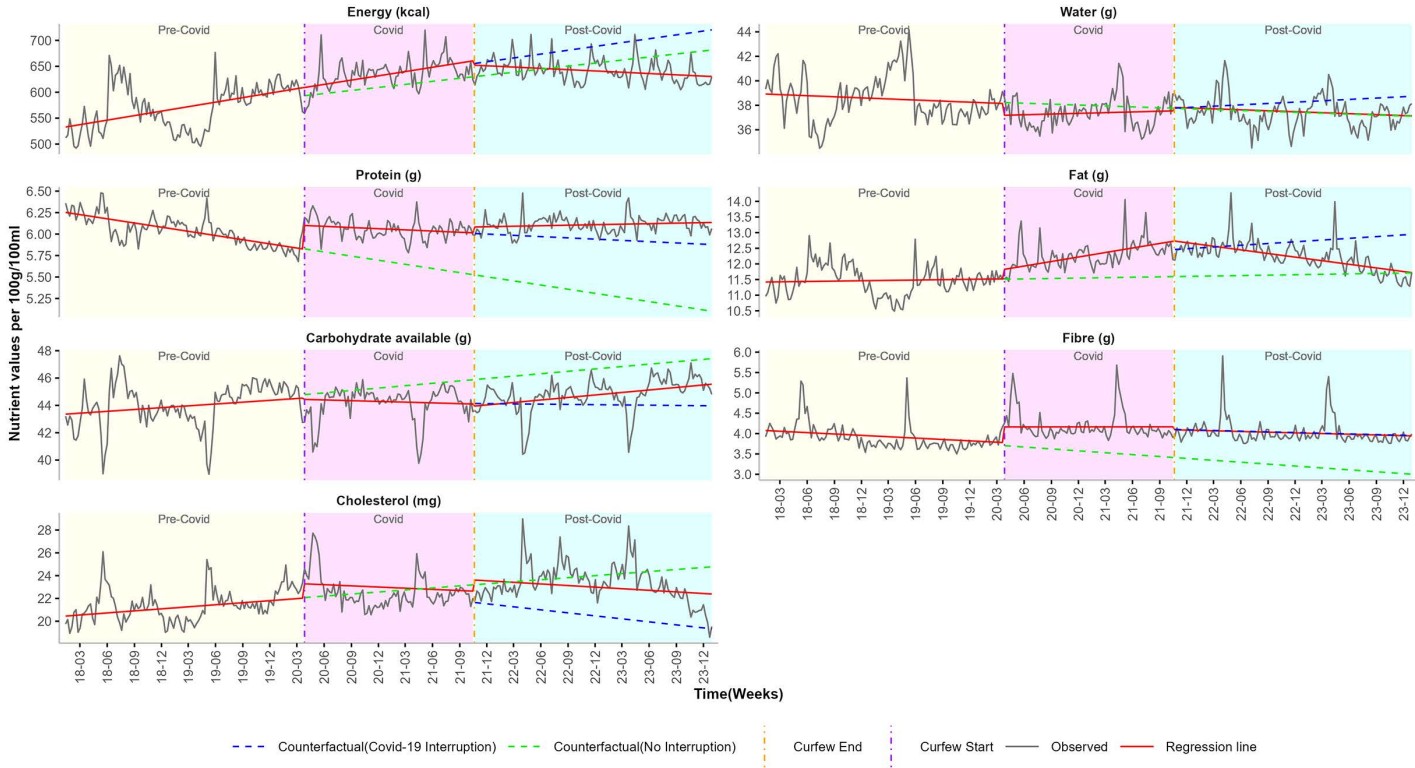

**Fig 2. ITS-GLS model weekly mean estimates and the counterfactual of proximate nutrients before, during and after pandemic restrictions.** Curfew start vertical line = 27th March 2020 (start of pandemic restrictions). Curfew end vertical line = 20th October 2021 (end of pandemic restrictions). Predictions were estimated in two parts: by extrapolating the pre-pandemic trend and by extrapolating the combined pre-pandemic and pandemic trend.

## Before COVID-19

The pre-COVID trend was characterized by declining purchases of processed culinary ingredients and unprocessed/minimally processed foods, alongside increasing trends in processed and ultra-processed food purchases. These findings are consistent with the growing body of evidence on the global nutritional transition, characterized by increasing reliance on industrially processed foods, driven by the expanded provision and aggressive marketing by food industries, as well as the rising affordability of ultra-processed products such as sugar-sweetened beverages [44,46]. Similar patterns have been empirically documented across multiple African contexts. Studies of urban food environments in South Africa and Ghana have shown increasing reliance on packaged and ultra-processed foods, particularly among middle-income households [49]. This reliance on industrially processed foods is associated with lower diet quality and a heightened risk of obesity, hypertension, Type 2 diabetes, and cardiovascular diseases [50–52]. The decline in culinary ingredients reflects a transition away from home cooking, possibly driven by time scarcity and urbanization [53].

Findings from the pre-COVID nutritional standpoint align with the growing body of literature indicating a transition towards a diet increasingly composed of energy-dense, nutrient-poor foods [45,46]. This is observed in the increases of energy and carbohydrates, paralleled by a decline in protein. Declines in essential micronutrients, i.e., calcium, iron, magnesium, potassium, sodium, and selenium, underscore the broader nutritional consequences of the transition away from whole foods. Reductions in vitamin A, vitamin B12, thiamin, niacin, and folate indicate a diminishing purchase of micronutrients critical to immune function, cognitive health, and cellular metabolism. Similar nutrient transitions, marked by

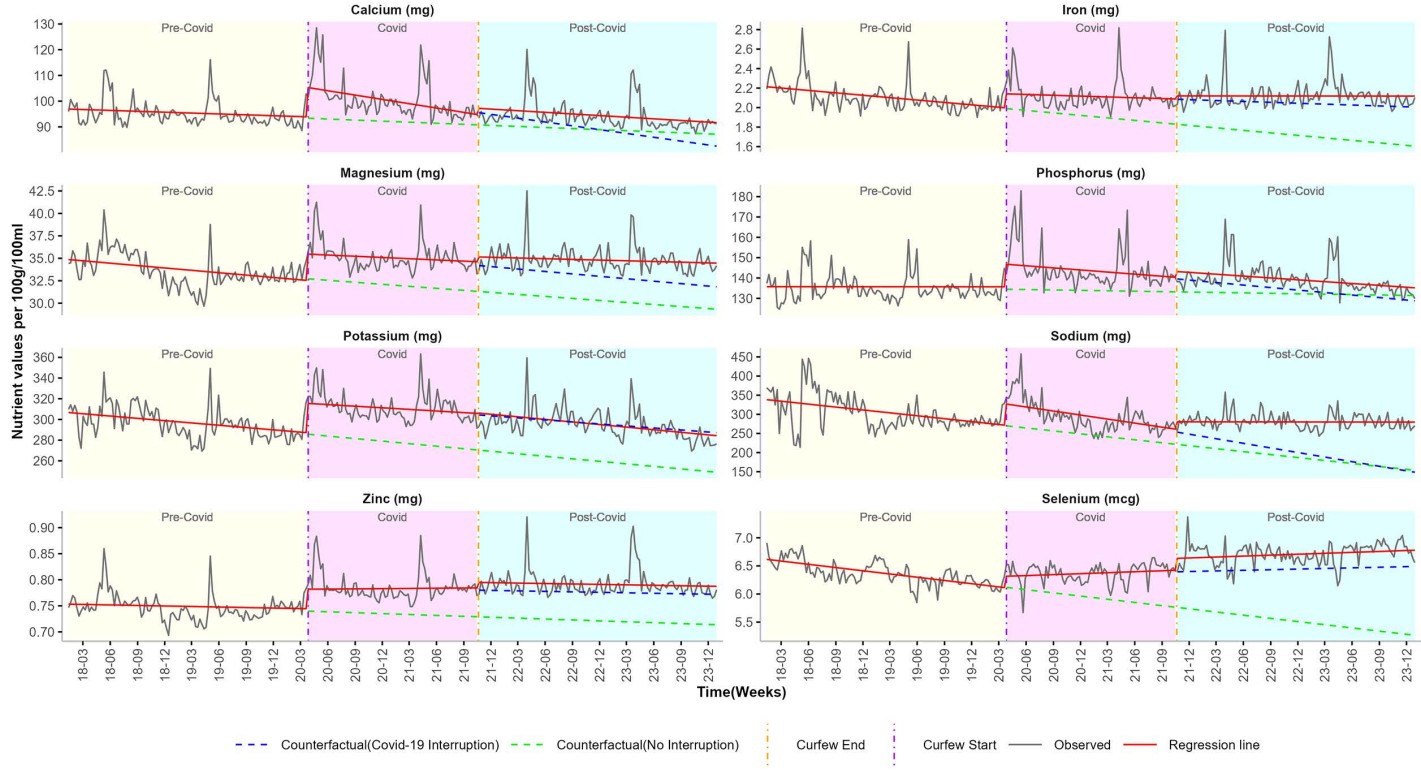

**Fig 3. ITS-GLS model weekly mean estimates and the counterfactual of mineral nutrients before, during and after pandemic restrictions.**
Curfew start vertical line = 27th March 2020 (start of pandemic restrictions). Curfew end vertical line = 20th October 2021 (end of pandemic restrictions). Predictions were estimated in two parts: by extrapolating the pre-pandemic trend and by extrapolating the combined pre-pandemic and pandemic trend.

rising energy and carbohydrate intake alongside declining protein and micronutrient density, have been reported in urban populations in Ethiopia and Nigeria [54]. Interestingly, vitamin C was the only micronutrient with an increasing trend, a pattern also observed in urban Ghanaian and South African settings [49], which may be attributable to purchases of vitamin C-fortified beverages or canned fruits to substitute purchases of fresh fruit.

### During COVID-19

Findings at the NOVA level reveal that the initial phase of lockdown was characterized by a sharp increase in purchases of unprocessed/minimally processed foods and processed foods. This finding suggests an immediate behavioral adjustment in food choices, possibly driven by restricted mobility, supply chain disruptions, and heightened health consciousness during the early lockdown period. The sharp decline in ultra-processed food purchases may indicate a preference for home cooking and the purchase of fresher ingredients as dining-out options diminished. Similar patterns from urban African settings suggest broadly comparable short-term disruptions, although with important contextual differences. Evidence from a multi-country study done in Tanzania, Ethiopia, Ghana, and Burkina Faso shows that during the early COVID-19 lockdown period, households increased purchases of whole grains, legumes, and green vegetables [27]. Similar findings from high-income settings were observed from an ITS study done in England using nationally representative panel data, which reported a 4.0% drop in ultra-processed food purchases [20] and studies in Italy that reported substantial increases in canned food purchases and a decline in the purchase of ultra-processed products [19,22]. The lack of sustained effects across all NOVA categories in our analysis indicates that while the lockdown temporarily altered the food environment,

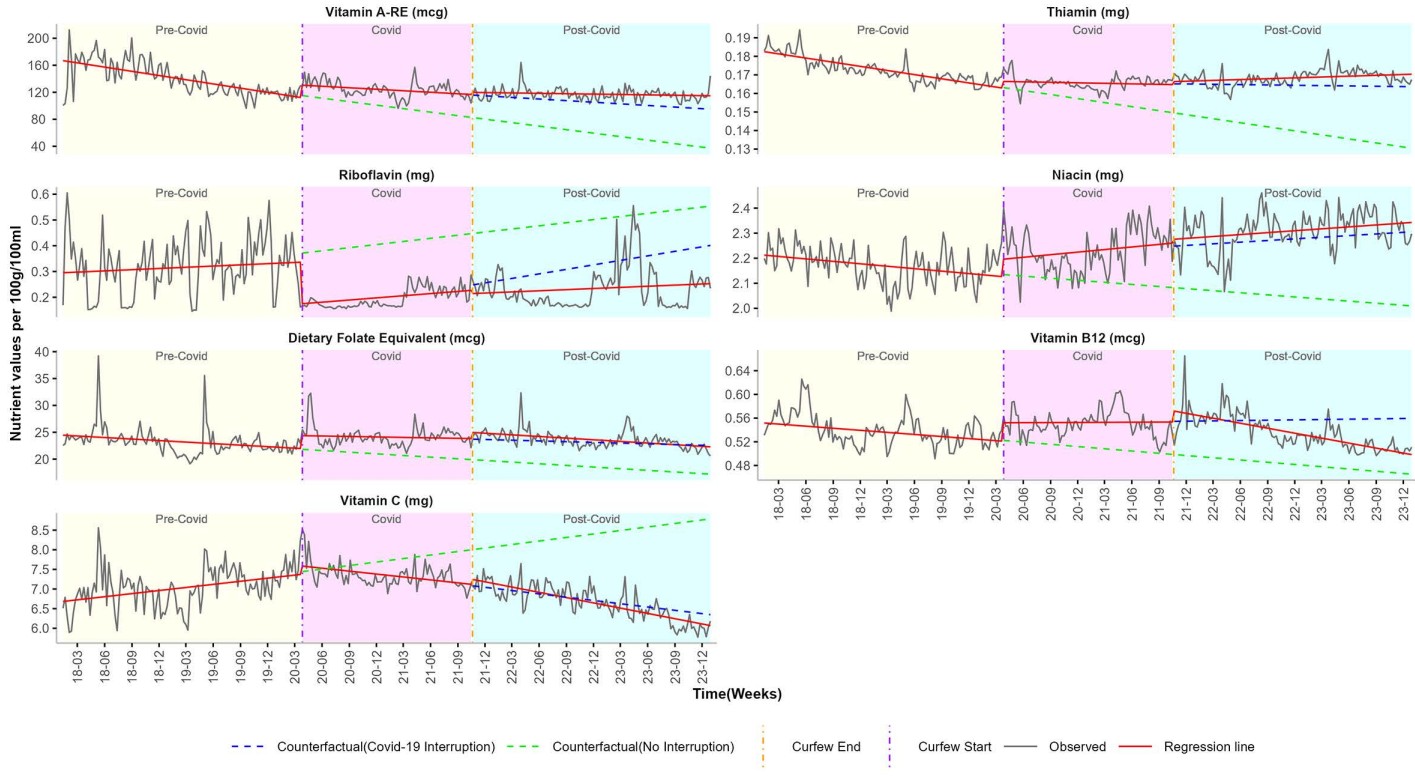

**Fig 4. ITS-GLS model weekly mean estimates and the counterfactual of vitamin nutrients before, during and after pandemic restrictions.** Curfew start vertical line = 27th March 2020 (start of pandemic restrictions). Curfew end vertical line = 20th October 2021 (end of pandemic restrictions). Predictions were estimated in two parts: by extrapolating the pre-pandemic trend and by extrapolating the combined pre-pandemic and pandemic trend.

many households reverted to pre-COVID purchasing habits as restrictions eased, likely facilitated by the rise of home deliveries through digital food retail platforms.

In terms of nutrition, the pandemic era was marked by selective improvements. Lockdown onset saw diet increases in fibre, protein, and multiple minerals (iron, magnesium, phosphorus, potassium, sodium, zinc, selenium, calcium), likely attributable to increased purchases of cereals, grain products, fish, seafood, meat, poultry, eggs, milk, dairy products, nuts, and seeds. Similar improvements in dietary diversity have been reported in urban households in Nigeria, Tanzania, Ethiopia, Ghana, and Burkina Faso, although these gains were often uneven and socioeconomically stratified [27]. Protein and selenium stood out for having both immediate and sustained increases, suggesting a potential dietary rebalancing toward purchases of meat, poultry, eggs, fish, seafood, nuts, and seeds. Notably, calcium spiked early but declined over time, potentially reflecting initial milk and dairy products stockpiling followed by normalisation. This trajectory warrants attention, given the role of calcium in bone health. However, carbohydrate declined slightly over time, and there were no changes in purchased energy, fat, water, or cholesterol, suggesting a modest recalibration in macronutrient balance rather than comprehensive dietary restructuring. At pandemic onset, there was an immediate increase in dietary folate, possibly driven by stockpiling behaviour early in the pandemic of non-perishable, shelf-stable fortified foods such as flour and breakfast cereals. In parallel, vitamin A, niacin, and vitamin B12 increased both immediately and in the long term, reflecting higher purchasing of animal-derived products. However, the pandemic decline in vitamin C and a sharp drop in riboflavin at lockdown onset suggest a potential deterioration in purchases of certain fruits, vegetables, and organ meats, a challenge widely reported across urban African studies due to disruptions in fresh food supply chains during lockdown

**Table 4. Model evaluation metrics from the full ITS-GLS and ARIMA models predicting the weekly proportion of NOVA classification and weekly mean nutrient values per 100g/100ml of food.**

| Variable | Category | Optimal ITS model | Train data (1–312 weeks) | | Test data (1–312 weeks) | | | |
|---|---|---|---|---|---|---|---|---|
| | | | AIC | BIC | RMSE | MAE | MAPE | MASE |
| NOVA food classification | Processed Culinary Ingredients | generalised least squares via corARMA(p=3, q=3) | -3210.5850 | -3161.9260 | 0.0018 | 0.0014 | 8.7904 | 1.1446 |
| | | ARIMA(5,0,0) errors | -3203.5988 | -3158.6828 | 0.0014 | 0.0010 | 6.7133 | 0.0670 |
| | Processed foods | generalised least squares via corARMA(p=2, q=2) ARIMA(1,0,0) errors | -2788.2223 | -2747.0492 | 0.0040 | 0.0028 | 10.5879 | 1.2943 |
| | | | -2776.9382 | -2746.9942 | 0.0028 | 0.0020 | 7.3690 | 0.0738 |
| | Ultra-processed foods | generalised least squares via corARMA(p=0, q=3) | -1773.2543 | -1735.8243 | 0.0211 | 0.0134 | 1.7533 | 1.4144 |
| | | ARIMA(0,0,3) errors | -1773.2543 | -1735.8243 | 0.0136 | 0.0082 | 1.0975 | 0.0107 |
| | Unprocessed/Minimally processed foods | generalised least squares via corARMA(p=0, q=3) | -1866.7029 | -1829.2729 | 0.0182 | 0.0119 | 6.1477 | 1.4519 |
| | | ARIMA(0,0,3) errors | -1866.7029 | -1829.2728 | 0.0117 | 0.0073 | 3.6337 | 0.0377 |
| Proximates | Carbohydrate available (g) | generalised least squares via corARMA(p=2, q=1) | 767.4460 | 804.8760 | 1.3028 | 0.9021 | 2.0392 | 1.4698 |
| | | ARIMA(3,0,2) errors | 778.5285 | 823.4446 | 0.8093 | 0.5528 | 1.2632 | 0.0125 |
| | Cholesterol (mg) | generalised least squares via corARMA(p=3, q=3) | 888.1360 | 936.7950 | 1.4790 | 1.1131 | 4.9759 | 1.4811 |
| | | ARIMA(1,0,1) errors | 891.2728 | 924.9598 | 0.9794 | 0.6720 | 2.9527 | 0.0300 |
| | Energy (kcal) | generalised least squares via corARMA(p=1, q=2) | 2844.3011 | 2881.7312 | 30.4596 | 21.4863 | 3.5926 | 1.2010 |
| | | ARIMA(1,0,0) errors | 2844.4412 | 2874.3852 | 22.4853 | 16.3111 | 2.6576 | 0.0266 |
| | Fat (g) | generalised least squares via corARMA(p=4, q=3) | 230.4887 | 282.8908 | 0.4186 | 0.2978 | 2.5095 | 1.0939 |
| | | ARIMA(1,0,2) errors | 238.8660 | 276.2961 | 0.3434 | 0.2421 | 1.9991 | 0.0202 |
| | Fibre (g) | generalised least squares via corARMA(p=3, q=3) | -26.0839 | 22.5751 | 0.3059 | 0.1884 | 4.6717 | 1.1729 |
| | | ARIMA(0,0,4) errors | -9.0197 | 32.1533 | 0.2300 | 0.1406 | 3.3223 | 0.0349 |
| | Protein (g) | generalised least squares via corARMA(p=1, q=0) | -605.1795 | -575.2354 | 0.1057 | 0.0807 | 1.3286 | 1.0560 |
| | | ARIMA(1,0,0) errors | -605.1795 | -575.2354 | 0.0894 | 0.0669 | 1.1004 | 0.0110 |
| | Water (g) | generalised least squares via corARMA(p=2, q=4) | 902.4497 | 951.1088 | 1.4866 | 1.1210 | 2.9531 | 1.3417 |
| | | ARIMA(5,0,0) errors | 906.7868 | 951.7028 | 0.9943 | 0.7478 | 1.9767 | 0.0198 |
| Minerals | Calcium (mg) | generalised least squares via corARMA(p=4, q=1) | 1794.4418 | 1839.3579 | 5.3506 | 3.5310 | 3.6442 | 1.1715 |
| | | ARIMA(0,0,4) errors | 1797.7756 | 1838.9486 | 4.1599 | 2.6474 | 2.6619 | 0.0275 |
| | Iron (mg) | generalised least squares via corARMA(p=3, q=2) | -487.4202 | -442.5042 | 0.1289 | 0.0836 | 3.9473 | 0.9402 |
| | | ARIMA(1,0,0) errors | -464.5311 | -434.5870 | 0.1120 | 0.0730 | 3.3603 | 0.0345 |
| | Magnesium (mg) | generalised least squares via corARMA(p=3, q=2) | 1022.1772 | 1067.0933 | 1.4978 | 1.0460 | 3.0364 | 1.0441 |
| | | ARIMA(4,0,0) errors | 1037.9828 | 1079.1558 | 1.2318 | 0.8237 | 2.3537 | 0.0239 |
| | Phosphorus (mg) | generalised least squares via corARMA(p=0, q=4) | 2030.4654 | 2071.6385 | 7.1252 | 4.7198 | 3.3850 | 1.0555 |
| | | ARIMA(0,0,4) errors | 2030.4655 | 2071.6385 | 6.0406 | 3.9275 | 2.7514 | 0.0282 |
| | Potassium (mg) | generalised least squares via corARMA(p=3, q=2) | 2393.4464 | 2438.3625 | 12.7048 | 9.2870 | 3.0925 | 1.0232 |
| | | ARIMA(0,0,5) errors | 2395.5261 | 2440.4421 | 10.8134 | 7.4521 | 2.4471 | 0.0248 |

*(Continued)*

**Table 4.** (Continued)

| Variable | Category | Optimal ITS model | Train data (1–312 weeks) | | Test data (1–312 weeks) | | | |
|---|---|---|---|---|---|---|---|---|
| | | | AIC | BIC | RMSE | MAE | MAPE | MASE |
| | Selenium (mcg) | generalised least squares via corARMA(p=4, q=4) | -296.4600 | -240.3150 | 0.1695 | 0.1321 | 2.0318 | 1.0985 |
| | | ARIMA(1,0,0) errors | -293.7302 | -263.7862 | 0.1472 | 0.1106 | 1.7113 | 0.0170 |
| | Sodium (mg) | generalised least squares via corARMA(p=4, q=3) | 2943.1464 | 2995.5484 | 31.6960 | 22.0312 | 7.3290 | 1.0190 |
| | | ARIMA(3,0,2) errors | 2946.0548 | 2990.9708 | 26.0507 | 18.3765 | 6.2065 | 0.0628 |
| | Zinc (mg) | generalised least squares via corARMA(p=3, q=2) | -1580.4114 | -1535.4954 | 0.0243 | 0.0158 | 2.0475 | 1.1190 |
| | | ARIMA(0,0,4) errors | -1562.7265 | -1521.5534 | 0.0191 | 0.0123 | 1.5677 | 0.0160 |
| Vitamins | Dietary Folate Equivalent (mcg) | generalised least squares via corARMA(p=0, q=4) | 1224.3806 | 1265.5536 | 1.9240 | 1.2031 | 5.0970 | 1.0119 |
| | | ARIMA(0,0,4) errors | 1224.3806 | 1265.5536 | 1.6608 | 0.9767 | 3.9621 | 0.0414 |
| | Niacin (mg) | generalised least squares via corARMA(p=2, q=1) | -809.4590 | -772.0289 | 0.0733 | 0.0586 | 2.6201 | 0.9950 |
| | | ARIMA(2,0,1) errors | -809.4589 | -772.0289 | 0.0640 | 0.0490 | 2.1994 | 0.0219 |
| | Riboflavin (mg) | generalised least squares via corARMA(p=4, q=4) | -794.6260 | -738.4809 | 0.0818 | 0.0595 | 22.4041 | 1.2927 |
| | | ARIMA(3,0,0) errors | -792.7715 | -755.3415 | 0.0657 | 0.0452 | 17.7673 | 0.1770 |
| | Thiamin (mg) | generalised least squares via corARMA(p=1, q=0) | -2711.2726 | -2681.3286 | 0.0038 | 0.0029 | 1.6934 | 1.1453 |
| | | ARIMA(3,0,0) errors | -2708.4631 | -2671.0331 | 0.0031 | 0.0023 | 1.3548 | 0.0136 |
| | Vitamin A-RE (mcg) | generalised least squares via corARMA(p=3, q=2) | 2430.1821 | 2475.0982 | 13.2481 | 9.2722 | 7.0484 | 0.9059 |
| | | ARIMA(4,0,0) errors | 2445.3760 | 2486.5490 | 11.7495 | 8.3465 | 6.4075 | 0.0656 |
| | Vitamin B12 (mcg) | generalised least squares via corARMA(p=4, q=3) | -1670.3771 | -1617.9750 | 0.0218 | 0.0159 | 2.9275 | 1.1911 |
| | | ARIMA(1,0,1) errors | -1644.1609 | -1610.4739 | 0.0168 | 0.0120 | 2.1902 | 0.0221 |
| | Vitamin C (mg) | generalised least squares via corARMA(p=3, q=3) | 182.6930 | 231.3521 | 0.3537 | 0.2617 | 3.7551 | 0.9373 |
| | | ARIMA(5,0,0) errors | 182.6462 | 227.5622 | 0.3118 | 0.2342 | 3.3504 | 0.0336 |

Note: AIC = Akaike Information Criterion; BIC = Bayesian Information Criterion; RMSE = Root Mean Squared Error; MAE = Mean Absolute Error; MAPE = Mean Absolute Percentage Error; MASE = Mean Absolute Scaled Error

[27]. In contrast, the steady increase in thiamin over time suggests a more gradual dietary adaptation to purchases of nuts, seeds, fish, seafood, and processed foods like fortified cereals, which became more prominent in urban food baskets during the pandemic [27].

### After COVID-19

In the NOVA classification, purchases of processed foods increased in the immediate post-lockdown period, followed by a gradual long-term decline. This pattern may reflect an initial rebound in convenience-oriented purchases, later offset by renewed health-conscious decisions or economic constraints limiting sustained demand for processed foods [47,48].

From a nutritional perspective, the sustained increase in carbohydrate and concurrent decline in fat and total energy may reflect increased purchases of lower-cost staple foods (e.g., grains) and reduced purchase of high-fat products

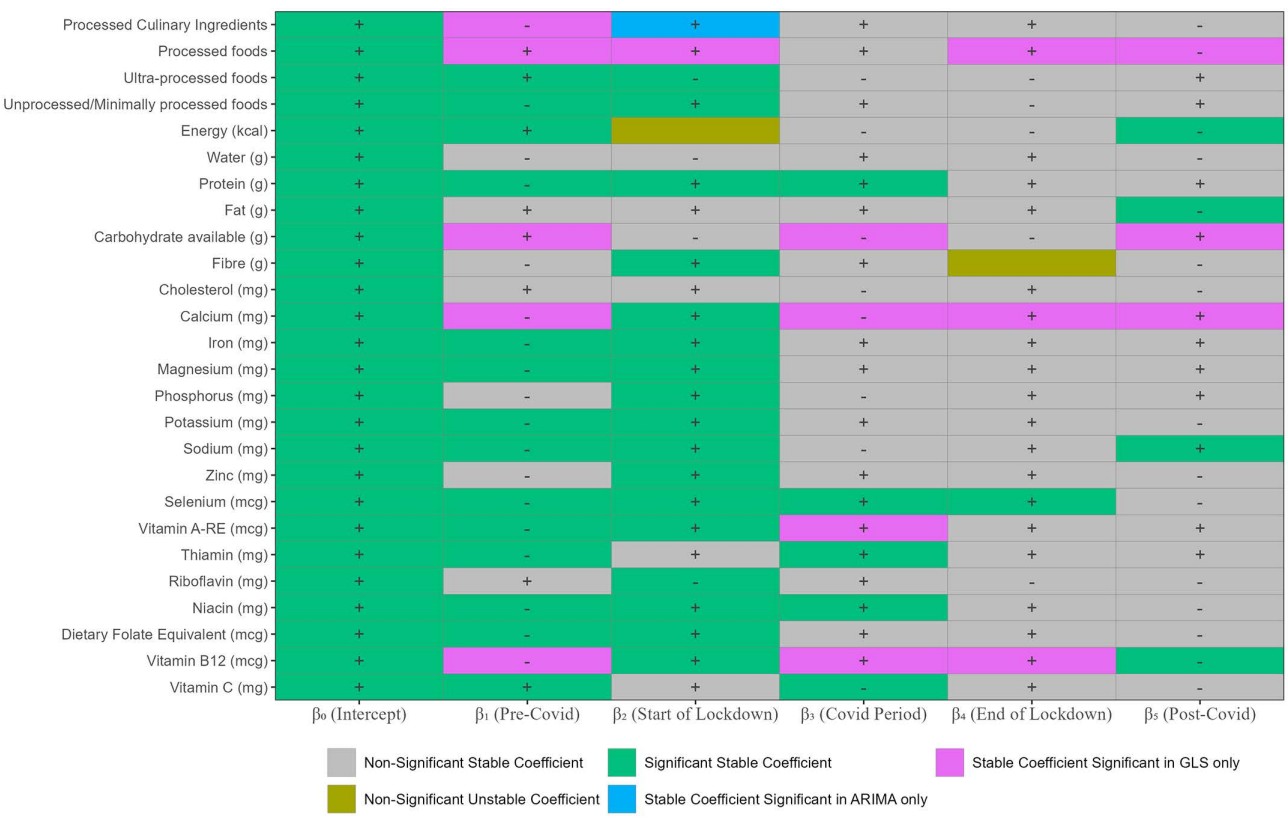

**Fig 5. Comparison of coefficients and p-values for full ITS GLS and ARIMA models.**

like savoury snacks. Micronutrient trends were largely stable. However, post-COVID increases in sodium and calcium suggest greater purchases of processed foods, dairy, or convenience foods rich in these nutrients. The increasing trend in sodium purchase in the long term is alarming, as it can increase the risk of hypertension and cardiovascular diseases [52]. The observed pattern in vitamin B12, of an initial post-lockdown increase followed by a long-term decline, may reflect changes in the availability or affordability of dairy products such as milk, cheese, and yogurt [47,48]. Such changes in nutrient composition can contribute to dietary monotony and raise concerns of both over-and under-nutrition in vulnerable populations.

## Policy implications

These findings present critical policy implications in the current Kenyan food environment. The continued high purchasing patterns of ultra-processed foods call for the adoption of the Kenya Nutrition Profile Model (KNPM), which provides guidance aimed at addressing the consumption of foods and beverages high in the nutrients of concern (sugars, fats, salts), hence promoting healthier food options [55]. The KNPM considers the development and implementation of food environment policies such as front-of-pack labeling, marketing restrictions, and fiscal policies. Front-of-pack labeling helps increase consumer awareness and encourages healthier purchasing choices. Interpretive labeling systems, such as warning labels, have been shown to enhance consumers' understanding of food purchases [56]. Regulating the marketing of unhealthy foods and beverages, especially to children and adolescents, is essential. Restricting advertising in school zones, during peak television viewing hours, and on online platforms aligns with WHO recommendations and may help

reduce exposure to unhealthy product promotions [57]. To counter the supermarket prominence of ultraprocessed foods, fiscal interventions such as taxes, tariff rates, and subsidies can be implemented to discourage the purchase of unhealthy foods while promoting the production and purchasing of healthier alternatives. These fiscal measures have been shown to contribute to the reduction in overweight and obesity in LMICs [58] and lower sugar intake in South Africa [59]. These policies, currently under consideration by the KNPM, are crucial for preventing further increases in nutrition-related non-communicable diseases (NR-NCDs) such as diabetes, cardiovascular diseases, and certain types of cancer.

### Strengths and limitations of the study

A key strength of our study lies in its originality and innovative use of granular grocery purchase data, combined with a robust quasi-experimental design. Given the current gap in studies leveraging large-scale data sources in Africa, this research offers a valuable benchmark for future studies that utilize unconventional data sources such as supermarket transaction records. These underexplored data sources can provide accurate measures for household or individual-level dietary behaviors, particularly during periods of crisis, such as pandemics. Secondly, unlike previous studies that have compared purchase patterns restricted to pre- and during-COVID-19 periods, our large sample size allowed us to split the data into pre-, during-, and post-COVID-19 phases, thereby providing additional insights into post-pandemic trends. Furthermore, we were able to link the grocery purchase data to food composition data, enabling the examination of changes in food purchasing patterns with detailed nutritional information.

Our study had some limitations. First, the study focused on supermarket transaction data, thus did not consider out-of-home purchases and alternative traditional food purchasing outlets such as kiosks, open-air markets, and street vendors. However, recent analyses indicate that off-trade purchases (i.e., supermarket transactions) account for the majority of food sales compared to on-site purchases. Second, the grocery data was drawn from one county (Nairobi) out of the 47 counties in Kenya; therefore, the data might not be representative of other regions in Kenya. However, Nairobi County is the most populous city that has the highest number of supermarkets and might represent the greatest population purchasing food from supermarkets. Third, the study focused on temporal changes in purchasing behaviour and did not adjust for temporal food price variation. Changes in prices during the COVID-19 period may have influenced purchasing behavior. Future analyses could incorporate time-varying price indices or unit price measures to better isolate behavioral effects from economic drivers. Finally, our data lacked sociodemographic characteristics of the shoppers, hence limited in multivariable modeling as we were not able to explore whether changes in food purchasing differed according to socio-economic status, region, gender, and age. Sociodemographic characteristics play a crucial role in shaping national food policies, as they allow for the development of targeted interventions by providing insights into how these factors influence purchase and dietary behaviors.

## 5. Conclusion and recommendations

Overall, clear contrasts emerge across the three periods, highlighting distinct and time-dependent behavioral dynamics. The pre-COVID period was characterized by concerning purchase patterns away from minimally processed, nutrient-rich foods and toward ultra-processed, energy-dense items. In contrast, the COVID-19 pandemic disrupted these existing purchasing trends, inducing short-term reactive improvements in several dimensions, including increased purchase of nutrient-dense staples and reductions in ultra-processed products. However, this temporary reversal was not maintained in the post-COVID period. Several pre-existing unhealthy purchasing patterns re-emerged, most notably the increase in purchases of ultra-processed foods, despite some sustained gains in select micronutrients. Taken together, the comparison across periods suggests that while the COVID-19 disruption temporarily altered purchasing behavior to more healthy patterns, these changes were largely transitory rather than transformational.

These findings have important public health implications, highlighting the need for sustained targeted public health interventions and food environment reforms, especially in the wake of crisis-induced behavior changes that could serve as

windows of opportunity for healthier food system transitions. In light of these, we recommend that policymakers implement measures to: 1) promote healthier food environments and reduce the purchase of ultra-processed foods, even beyond the context of emergency response; 2) include support for nutrient-dense food access and education during future public health emergencies; and 3) improve the affordability and availability of nutritious foods, particularly for low-income households, while addressing the broader structural determinants of food choice and access.

## Supporting information

**S1 Appendix. Classified food items in the data according to NOVA.**
(DOCX)

**S1 Table. Parameter estimates, confidence intervals, and Z-test p-values from the full ITS-ARIMA models predicting the weekly proportion of NOVA classification.**
(DOCX)

**S2 Table. Parameter estimates, confidence intervals, and Z-test p-values from the full ITS-ARIMA models predicting the weekly mean nutrient values per 100g/100ml of food.**
(DOCX)

**S3 Table. Parameter estimates, confidence intervals, and Z-test p-values from the pre-pandemic ITS-GLS models predicting the weekly proportion of NOVA classification and weekly mean nutrient values per 100g/100ml of food.**
(DOCX)

**S4 Table. Parameter estimates, confidence intervals, and Z-test p-values from the pre-pandemic ITS-ARIMA models predicting the weekly proportion of NOVA classification and weekly mean nutrient values per 100g/100ml of food.**
(DOCX)

**S5 Table. Model evaluation metrics from the pre-pandemic ITS-GLS and ARIMA models predicting the weekly proportion of NOVA classification and the weekly mean nutrient values per 100g/100ml of food.**
(DOCX)

**S6 Table. Parameter estimates, confidence intervals, and Z-test p-values from the combined pre-pandemic and pandemic ITS-GLS models predicting the weekly proportion of NOVA classification and weekly mean nutrient values per 100g/100ml of food.**
(DOCX)

**S7 Table. Parameter estimates, confidence intervals, and Z-test p-values from the combined pre-pandemic and pandemic ITS-ARIMA models predicting the weekly proportion of NOVA classification and weekly mean nutrient values per 100g/100ml of food.**
(DOCX)

**S8 Table. Model evaluation metrics from the combined pre-pandemic and pandemic ITS-GLS and ARIMA models predicting the weekly proportion of NOVA classification and the weekly mean nutrient values per 100g/100ml of food.**
(DOCX)

**S1 Fig. Weekly proportions and ITS-GLS model predictions of NOVA food classification before, during, and after pandemic restrictions.** Curfew start vertical line = 27th March 2020 (start of pandemic restrictions). Curfew end vertical

line = 20th October 2021 (end of pandemic restrictions). Predictions were estimated in two parts: by extrapolating the pre-pandemic trend and by extrapolating the combined pre-pandemic and pandemic trend.
(TIFF)

**S2 Fig. ITS-GLS model weekly mean estimates and the predictions of proximate nutrients before, during, and after pandemic restrictions.** Curfew start vertical line = 27th March 2020 (start of pandemic restrictions). Curfew end vertical line = 20th October 2021 (end of pandemic restrictions). Predictions were estimated in two parts: by extrapolating the pre-pandemic trend and by extrapolating the combined pre-pandemic and pandemic trend.
(TIFF)

**S3 Fig. ITS-GLS model weekly mean estimates and the predictions of mineral nutrients before, during, and after pandemic restrictions.** Curfew start vertical line = 27th March 2020 (start of pandemic restrictions). Curfew end vertical line = 20th October 2021 (end of pandemic restrictions). Predictions were estimated in two parts: by extrapolating the pre-pandemic trend and by extrapolating the combined pre-pandemic and pandemic trend.
(TIFF)

**S4 Fig. ITS-GLS model weekly mean estimates and the predictions of vitamin nutrients before, during, and after pandemic restrictions.** Curfew start vertical line = 27th March 2020 (start of pandemic restrictions). Curfew end vertical line = 20th October 2021 (end of pandemic restrictions). Predictions were estimated in two parts: by extrapolating the pre-pandemic trend and by extrapolating the combined pre-pandemic and pandemic trend.
(TIFF)

## Acknowledgments

The authors would like to thank the supermarkets for providing the data.

## Author contributions

**Conceptualization:** Reinpeter Momanyi, Tatenda Duncan Kavu.

**Data curation:** Reinpeter Momanyi.

**Formal analysis:** Reinpeter Momanyi.

**Funding acquisition:** Steve Cygu, Agnes Kiragga.

**Methodology:** Reinpeter Momanyi, Tatenda Duncan Kavu, Daniel Mtai Mwanga, Caroline H. Karugu, Gershim Asiki.

**Project administration:** Steve Cygu.

**Supervision:** Tatenda Duncan Kavu, Agnes Kiragga.

**Validation:** Reinpeter Momanyi.

**Visualization:** Reinpeter Momanyi.

**Writing – original draft:** Reinpeter Momanyi.

**Writing – review & editing:** Reinpeter Momanyi, Tatenda Duncan Kavu, Daniel Mtai Mwanga, Caroline H. Karugu, Steve Cygu, Gershim Asiki, Agnes Kiragga.

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
