## [Decision Letter · Decision Letter 0]

22 Feb 2026

PGPH-D-25-03211

Food Purchase Patterns In Nairobi before, during, and after the COVID-19 Pandemic lockdown measures

Dear Dr. Momanyi,

Thank you for submitting your manuscript to PLOS Global Public Health. After careful consideration, we feel that it has merit but does not fully meet PLOS Global Public Health’s publication criteria as it currently stands. Therefore, we invite you to submit a revised version of the manuscript that addresses the points raised during the review process.

We look forward to receiving your revised manuscript.

Kind regards,

Hasanain Faisal Ghazi, phd

Academic Editor

Journal Requirements:

i. Please clarify all sources of financial support for your study. List the grants, grant numbers, and organizations that funded your study, including funding received from your institution. Please note that suppliers of material support, including research materials, should be recognized in the Acknowledgements section rather than in the Financial Disclosure.

ii. State the initials, alongside each funding source, of each author to receive each grant. For example: "This work was supported by the National Institutes of Health (####### to AM; ###### to CJ) and the National Science Foundation (###### to AM)."

iii. State what role the funders took in the study. If the funders had no role in your study, please state: “The funders had no role in study design, data collection and analysis, decision to publish, or preparation of the manuscript.”

iv. If any authors received a salary from any of your funders, please state which authors and which funders.

2. Please ensure that your Ethics Statement is available in its entirety at the beginning of your Methods section, under a subheading 'Ethics Statement'.

3. For studies involving third-party data, we encourage authors to share any data specific to their analyses that they can legally distribute. PLOS recognizes, however, that authors may be using third-party data they do not have the rights to share. When third-party data cannot be publicly shared, authors must provide all information necessary for interested researchers to apply to gain access to the data. (https://journals.plos.org/plosone/s/data-availability#loc-acceptable-data-access-restrictions)

Additional Editor Comments (if provided):

please respond to reviewers' comments

Reviewers' comments:

Reviewer's Responses to Questions

**Comments to the Author**

1. Does this manuscript meet PLOS Global Public Health’s publication criteria? Is the manuscript technically sound, and do the data support the conclusions? The manuscript must describe methodologically and ethically rigorous research with conclusions that are appropriately drawn based on the data presented.

Reviewer #1: Yes

Reviewer #2: Yes

Reviewer #3: Yes

Reviewer #4: Yes

2. Has the statistical analysis been performed appropriately and rigorously?

Reviewer #1: Yes

Reviewer #2: I don't know

Reviewer #3: Yes

Reviewer #4: Yes

3. Have the authors made all data underlying the findings in their manuscript fully available (please refer to the Data Availability Statement at the start of the manuscript PDF file)?

Reviewer #1: Yes

Reviewer #2: Yes

Reviewer #3: Yes

Reviewer #4: Yes

4. Is the manuscript presented in an intelligible fashion and written in standard English?

Reviewer #1: Yes

Reviewer #2: Yes

Reviewer #3: Yes

Reviewer #4: Yes

5. Review Comments to the Author

Reviewer #1: Key recommendations:

ESSENTIAL REVISIONS:

Systematically replace "intake," "consumption," and "dietary" terminology with "purchased" or "purchasing" throughout the manuscript. This is critical since the study measured purchases, not consumption.

Address multiple testing: With 162 statistical tests, consider applying corrections (Bonferroni, FDR) or explicitly acknowledge this limitation and emphasize effect sizes.

Expand discussion of generalizability limitations, particularly regarding socioeconomic bias in supermarket shoppers versus Nairobi's broader population.

Better contextualize pre-COVID trends with comparative data from similar urban African settings.

STRONGLY RECOMMENDED:

Add discussion distinguishing statistically significant from practically meaningful effects

Provide more specific COVID-19 context for Nairobi (case rates, economic indicators, food prices)

Clarify NOVA classification procedures and quality control measures

Improve figure accessibility (larger fonts, colorblind-friendly colors)

SUGGESTED:

Justify weekly temporal aggregation choice

Expand analysis separating stockpiling from sustained behavioral change

Provide model selection details for ARMA orders

Translate key coefficients into interpretable effect sizes

This is valuable work addressing an important gap in African food systems research with innovative data and rigorous methods. With the revisions outlined above, it will make a strong contribution to the literature on food purchasing behavior during public health crises.

Reviewer #2: The study has brought some interesting finding, however, it lack to explain the significance of the study on the introduction section which will provide more attention to the readers.

How was price factors has anlayzed on the food pattern consumption? It would be better to highlight if analysis has been done.

Reviewer #3: The researchers have used the available data and statistical tools to understand and explain the research. It follow rigorous scientific method for the ETL and data analysis. I didn't find significant concern.

Reviewer #4: The manuscript on Food purchase patterns in Nairobi before, during, and after the COVID-19 pandemic lockdown measures is relevant and appropriate for the journal. The authors should see the attachment for the necessary comments.

6. PLOS authors have the option to publish the peer review history of their article (what does this mean?). If published, this will include your full peer review and any attached files.

**Do you want your identity to be public for this peer review?** For information about this choice, including consent withdrawal, please see our Privacy Policy.

Reviewer #1: No

Reviewer #2: No

Reviewer #3: **Yes:** Hailegiorgis Yirgu

Reviewer #4: No

Figure Resubmissions:

---

## [Editor Report · Decision Letter 1]

12 May 2026

Food purchase patterns in Nairobi before, during, and after the COVID-19 pandemic lockdown measures

PGPH-D-25-03211R1

Dear Mr Momanyi,

We are pleased to inform you that your manuscript 'Food purchase patterns in Nairobi before, during, and after the COVID-19 pandemic lockdown measures' has been provisionally accepted for publication in PLOS Global Public Health.

Best regards,

Hasanain Faisal Ghazi, phd

Academic Editor